# Handling Missing Responses under Cluster Dependence with Applications to Language Model Evaluation

**Zhenghao Zeng**
Stanford University
zhzeng@stanford.edu

**David Arbour**
Adobe Research
arbour@adobe.com

**Avi Feller**
University of California, Berkeley
afeller@berkeley.edu

**Ishita Dasgupta**
Adobe Research
idasgupt@adobe.com

**Atanu R Sinha**
Adobe Research
atr@adobe.com

**Edward H. Kennedy**
Carnegie Mellon University
edward@stat.cmu.edu

## Abstract

Human annotations play a crucial role in evaluating the performance of GenAI models. Two common challenges in practice, however, are missing annotations (the response variable of interest) and cluster dependence among human-AI interactions (e.g., questions asked by the same user may be highly correlated). Reliable inference must address both issues to achieve unbiased estimation and appropriately quantify uncertainty when estimating average scores from human annotations. In this paper, we analyze the doubly robust estimator, a widely used method in missing data analysis and causal inference, applied to this setting and establish novel theoretical properties under cluster dependence. We further illustrate our findings through simulations and a real-world conversation quality dataset. Our theoretical and empirical results underscore the importance of incorporating cluster dependence in missing response problems to perform valid statistical inference.

## 1 Introduction

Missing response/outcome variables are common in empirical research and present many challenges to data analysis and interpretation. Such missingness can occur for various reasons, including nonresponse in surveys (Hansen and Hurwitz, 1946; Chen and Haziza, 2019), dropout in longitudinal studies (Hogan et al., 2004), or data entry errors (Bound et al., 2001; Schennach, 2016). If not properly addressed, missingness can lead to biased estimates, reduced statistical power, and invalid conclusions. Researchers have developed methods that leverage observed data to estimate missing values under assumptions about the missingness mechanism, typically assuming i.i.d sampling. Examples common in causal inference include outcome modeling, re-weighting, and combinations of the two (Robins et al., 1994; Little and Rubin, 2019).

Clustered data, commonly encountered in fields such as education (Lüdtke et al., 2011), healthcare (Austin and Merlo, 2017), and the social sciences (McNeish and Stapleton, 2016), refers to data in which observations are naturally grouped into clusters or hierarchies. Typical examples include students nested within schools or patients nested within hospitals, where clusters (e.g., schools or hospitals) are sampled first, followed by individuals within those clusters. Another form of clustered sampling arises in settings with repeated measurements on the same individuals. For instance, in the evaluation of large language models (LLMs), users often provide multi-turn feedback, with user-system interactions generated sequentially for each user. In this context, different users can be treated as separate clusters. This clustering introduces within-cluster correlation, violating the

assumption of independence commonly required by standard statistical methods. Researchers often account for such clustering using specialized techniques such as multilevel modeling (Raudenbush and Bryk, 2002), generalized estimating equations (Zorn, 2001), and cluster-robust inference (Hansen and Lee, 2019). These methods provide valid inference by incorporating within-cluster variability and appropriately accounting for the hierarchical structure of the data.

Analyzing clustered data while addressing missing responses and conducting causal inference with individual-level treatments introduces additional complexities. Yang (2018) proposed a calibration technique that balances both observed individual-level confounders and unobserved cluster-level confounders. Suk et al. (2021); Suk and Kang (2022) adapted modern machine learning methods, such as causal forests (Wager and Athey, 2018), to multilevel observational data. Park and Kang (2021) introduced a refined method that models the conditional propensity score and the outcome covariance structure to account for within-cluster correlations in the estimation procedures. For a comprehensive review and comparison of propensity weighting approaches in multilevel data, see Fuentes et al. (2022); Chang and Stuart (2022).

In this work, we study the properties of the widely used doubly robust estimator for handling missing outcomes in clustered data. In the i.i.d. setting, this estimator is consistent for the average outcome when either the outcome regression or the propensity score is correctly specified, and it achieves parametric convergence rates even when nuisance functions are estimated at slower, nonparametric rates. However, its behavior under cluster dependence is less well understood. Extending recent work from Park and Kang (2021), we first establish a novel form of asymptotic normality for the doubly robust estimator in the presence of clustered data by leveraging recent central limit theorems designed for such settings. We show that the convergence rate depends on both the within-cluster correlation of individual influence functions and the error in estimating nuisance functions. Notably, the estimator can achieve faster rates than the conventional $\sqrt{G}$-rate ($G$ is the number of clusters) when cluster sizes are large and within-cluster dependence is weak. We then conduct extensive simulation studies, which highlight the importance of accounting for within-cluster correlation and using a cluster-robust variance estimator to obtain valid inference. Our results provide theoretical justification for using doubly robust estimators in the analysis of clustered data, especially when the cluster sizes are unbounded. The proposed methods, such as incorporating summaries of historical information into the estimation procedure, have important applications in multi-turn LLM evaluation with missing human annotations, as demonstrated in our real-data example.

The remainder of this paper is organized as follows: Section 2 introduces the problem setup and notation. Section 3 examines the properties of the doubly robust estimator under homogeneous sampling for clustered data. Our results extend the theoretical analysis in Park and Kang (2021) by allowing for unbounded cluster sizes and rates adaptive to cluster dependence. Section 4 presents a method for incorporating temporal dependence within each cluster into estimation with interesting applications in LLM evaluations. Numerical experiments and a real-world example that illustrate our results are provided in Section 5–6. Finally, we conclude with additional discussion in Section 7. All proofs, additional discussion on related work and details of numerical experiments are included in the Appendix.

## 2   Setup and Notation

Let $g \in [G]$ denote the index of $G$ clusters and $i \in [n_g]$ index $n_g$ individuals in the $g$-th cluster. For each individual $i$ in the $g$-th cluster, let $\mathbf{W}_{gi}$ represent the individual-level covariates and $\mathbf{X}_g$ represent the cluster-level covariates. For example, in educational assessment studies, clusters typically correspond to different schools, with individuals being the students within those schools. Cluster-level covariates might include the type and location of the school, while individual-level covariates could encompass factors such as age, test scores, and prior educational experience of students. In the context of LLM evaluation, one user (associated with user-level covariates $\mathbf{X}_g$) typically asks multiple questions and provides feedback. Different questions and their corresponding answers (i.e., the individual-level covariates $\mathbf{W}_{gi}$) generated by the LLM are often correlated. Instead of treating all question-answer pairs as independent data, it may be more appropriate to consider questions and answers associated with the same user as a cluster, where data from different clusters are independent, but dependencies within clusters exist.

Let $Y_{gi}$ denote the outcome of interest for $i$-th individual in $g$-th cluster. In education assessment, $Y_{gi}$ may represent the score of a student's academic performance or psychological well-being. In LLM evaluations, $Y_{gi}$ is the score provided by the user and we are interested in estimating the average score to understand the performance of the LLM system/platform. In real applications, the surveyed outcome $Y_{gi}$ may not be observed for all data points. For instance, in the aforementioned examples, some students or users may choose not to provide their scores, resulting in missing response data. Let $R_{gi}$ denote the missing indicator, where $R_{gi} = 1$ if $Y_{gi}$ is observed and $R_{gi} = 0$ otherwise. With this notation, the observed data of each individual is $\mathbf{O}_{gi} = (\mathbf{X}_g, \mathbf{W}_{gi}, R_{gi}, R_{gi}Y_{gi})$.

## 3 Clustered Missing Data under Homogeneous Sampling

In this section, we begin by considering a simplified setting where the observed data $\{\mathbf{O}_{gi}, 1 \leq i \leq n_g, 1 \leq g \leq G\}$ are assumed to be identically distributed, and the missingness of each individual's outcome is solely dependent on their own covariates. The analysis in this homogeneous sampling setting extends naturally from the i.i.d. case. To identify the average outcome in the missing data setting, we impose the Missing at Random (MAR) assumption on the data-generating process.

**Assumption 1** (Missing at random). $R_{gi} \perp\!\!\!\perp Y_{gi} \mid \mathbf{X}_g, \mathbf{W}_{gi}$ *and* $\pi(\mathbf{X}_g, \mathbf{W}_{gi}) := \mathbb{P}(R_{gi} = 1 \mid \mathbf{X}_g, \mathbf{W}_{gi}) > 0$ *almost surely.*

Assumption 1 requires that the cluster-level covariates and individual-level covariates together fully explain the missingness mechanism. When unobserved confounders may influence the missingness mechanism, the MAR assumption may no longer hold. To assess the sensitivity of our results to such violations, we can follow the framework of Cinelli and Hazlett (2020), which extends classical omitted variable bias analysis. This approach quantifies the strength that an unobserved confounder would need to exhibit—measured by its partial $R^2$ with both the treatment or missingness indicator and the outcome—to reduce the estimated effect to zero or render it statistically insignificant. The robustness value (RV) summarizes this threshold and can be benchmarked against observed covariates for interpretation.

Assumption 1 also requires the missingness mechanism to be *single-level* and homogeneous across clusters. In scenarios where the missingness mechanism is known to be heterogeneous and the mean outcome within each cluster is of interest (i.e., $\mathbb{E}[Y_{gi} \mid G = g]$), researchers can build cluster-specific propensity score models (e.g., random fixed-effects logistic models) to achieve better balance within clusters (Li et al., 2013; Thoemmes and West, 2011; Arpino and Mealli, 2011). The trade-off is that such an approach requires sufficiently large cluster sizes to reliably estimate propensity scores for each cluster. In our approach, propensity scores estimated from single-level models can be effectively used to balance observed covariates across clusters and to estimate the average outcome over all individuals (Suk et al., 2021; Park and Kang, 2021).

Consider the following data-generating process: First, sample $G$ i.i.d. cluster-level covariates $\mathbf{X}_1, \ldots, \mathbf{X}_G \sim \mathbb{P}_{\mathbf{X}}$. Within each cluster, $n_g$ identically distributed (but typically not independent due to within-cluster dependency) individual-level covariates $\mathbf{W}_{g1}, \ldots, \mathbf{W}_{gn_g} \sim \mathbb{P}_{\mathbf{W}|\mathbf{X}_g}$ are sampled. For each individual, the missing indicator $R_{gi}$ is then generated from $\mathrm{Bernoulli}(\pi(\mathbf{X}_g, \mathbf{W}_{gi}))$, followed by sampling $Y_{gi}$ from $\mathbb{P}_{Y|\mathbf{X}_g, \mathbf{W}_{gi}, R_{gi}=1}$ with conditional mean $\mu(\mathbf{X}_g, \mathbf{W}_{gi})$. Note that we assume the regression function $\mathbb{E}[R_{gi}Y_{gi} \mid \mathbf{X}_g, \mathbf{W}_{gi}, R_{gi} = 1] = \mu(\mathbf{X}_g, \mathbf{W}_{gi})$, implying it is also not cluster-specific. Under this sampling scheme, the observations $\{\mathbf{O}_{gi} = (\mathbf{X}_g, \mathbf{W}_{gi}, R_{gi}, R_{gi}Y_{gi}), 1 \leq i \leq n_g, 1 \leq g \leq G\}$ are identically distributed and $\{\mathbf{O}_{gi}, 1 \leq i \leq n_g\}$ are independent of $\{\mathbf{O}_{hj}, 1 \leq j \leq n_h\}$ for $g \neq h$ (i.e., the clusters are independent). However within the cluster, the dependency among $\{\mathbf{W}_{gi}, 1 \leq i \leq n_g\}$ is arbitrary. The likelihood function of $\{\mathbf{O}_{gi} = (\mathbf{X}_g, \mathbf{W}_{gi}, R_{gi}, R_{gi}Y_{gi}), 1 \leq i \leq n_g, 1 \leq g \leq G\}$ is

$$
\prod_{g=1}^{G} \Big\{ f_{\boldsymbol{x}}(\mathbf{X}_g) f_{\boldsymbol{w}}(\mathbf{W}_{g1}, \ldots, \mathbf{W}_{gn_g} \mid \mathbf{X}_g)
$$
$$
\times \prod_{i=1}^{n_g} [f_y(Y_{gi} \mid \mathbf{X}_g, \mathbf{W}_{gi}, R_{gi} = 1) \pi(\mathbf{X}_g, \mathbf{W}_{gi})]^{R_{gi}} (1 - \pi(\mathbf{X}_g, \mathbf{W}_{gi}))^{1-R_{gi}} \Big\}.
$$

When $\mathbf{X}_g$ fully explains the dependence among $\mathbf{W}_1, \ldots, \mathbf{W}_{n_g}$, we can express their joint distribution as $f_{\boldsymbol{w}}(\mathbf{W}_1, \ldots, \mathbf{W}_{n_g} \mid \mathbf{X}_g) = \prod_{i=1}^{n_g} f_{\boldsymbol{w}}(\mathbf{W}_i \mid \mathbf{X}_g)$, i.e., the individual-level covariates

$\mathbf{W}_1, \ldots, \mathbf{W}_{n_g}$ are conditionally independent given the cluster-level covariates $\mathbf{X}_g$. However, we do not impose this assumption on the data-generating process for generality. In the extreme case, it is possible that $\mathbf{O}_{g1} = \cdots = \mathbf{O}_{gn_g}$, meaning all observations within the $g$-th cluster are identical. Let $n = \sum_{g=1}^{G} n_g$ denote the total sample size. In this work, we allow $n_g \to \infty$ as $n \to \infty$.

In this section, we are interested in estimating the average outcome $\theta = \mathbb{E}[Y_{gi}]$ across all individuals, where each individual is given equal weight. Under Assumption 1, $\mathbb{E}[Y_{gi}]$ is identified as

$$\theta = \mathbb{E}[Y_{gi}] = \mathbb{E}[\mathbb{E}(R_{gi}Y_{gi} \mid \mathbf{X}_g, \mathbf{W}_{gi}, R_{gi} = 1)] = \mathbb{E}[\mu(\mathbf{X}_g, \mathbf{W}_{gi})]. \tag{1}$$

Since the distribution of $(\mathbf{X}_g, \mathbf{W}_{gi})$ is consistent across all $g$ and $i$, $\theta$ is independent of both $g$ and $i$, ensuring that it is well-defined.

## 3.1 Doubly Robust Estimation

Given expression (1) and an estimator of $\mu$ as $\hat{\mu}$, a natural plug-in-style estimator is

$$\hat{\theta}_{\mathrm{OR}} = \frac{1}{n} \sum_{g=1}^{G} \sum_{i=1}^{n_g} \hat{\mu}(\mathbf{X}_g, \mathbf{W}_{gi}).$$

This estimator corresponds to regression-based imputation and is consistent (under mild conditions) when $\hat{\mu}$ is consistent. However, the plug-in-style estimator usually suffers from first-order bias and is not robust to model misspecification (Bang and Robins, 2005; Funk et al., 2011). To address these issues, we consider the following doubly robust estimator that leverages both the estimated outcome model $\hat{\mu}$ and propensity score $\hat{\pi}$ (Robins et al., 1994; Scharfstein et al., 1999; Kennedy, 2024):

$$\hat{\theta}_{\mathrm{DR}} = \frac{1}{n} \sum_{g=1}^{G} \sum_{i=1}^{n_g} \left[ \frac{R_{gi}(Y_{gi} - \hat{\mu}(\mathbf{X}_g, \mathbf{W}_{gi}))}{\hat{\pi}(\mathbf{X}_g, \mathbf{W}_{gi})} + \hat{\mu}(\mathbf{X}_g, \mathbf{W}_{gi}) \right]. \tag{2}$$

In the classic i.i.d. setting, the doubly robust estimator remains consistent as long as either $\hat{\mu}$ or $\hat{\pi}$ is consistent, with conditional bias depending on the product of nuisance estimation errors. The following theorem characterizes its similar theoretical guarantees in the clustered setting.

**Theorem 1.** *Let* $\varphi(\mathbf{O}_{gi}) = \frac{R_{gi}(Y_{gi} - \mu(\mathbf{X}_g, \mathbf{W}_{gi}))}{\pi(\mathbf{X}_g, \mathbf{W}_{gi})} + \mu(\mathbf{X}_g, \mathbf{W}_{gi})$ *be the individual influence function. Under Assumption 1, assume there exist some constant* $C, c > 0$ *such that*

*1. For some* $r \geq 2$*, we have*

$$\mathbb{E}\left[|\varphi(\mathbf{O}_{gi})|^r\right] < \infty, \quad \frac{\left(\sum_{g=1}^{G} n_g^r\right)^{2/r}}{n} \leq C < \infty, \quad \max_{g \leq G} \frac{n_g^2}{n} \to 0. \tag{3}$$

*2.* $\Omega_n = \frac{1}{n} \sum_{g=1}^{G} \mathrm{Var}\left(\sum_{i=1}^{n_g} \varphi(\mathbf{O}_{gi})\right) \geq c > 0.$

*3.* $\hat{\pi}, \hat{\mu}$ *are estimated from a separate independent sample* $D$ *satisfying* $\hat{\pi} \geq c > 0$.

*Then we have*

$$\hat{\theta}_{DR} - \theta = \frac{1}{n} \sum_{g=1}^{G} \sum_{i=1}^{n_g} (\varphi(\mathbf{O}_{gi}) - \theta) + R_1 + R_2,$$

$$R_1 = O_{\mathbb{P}}\left(\|\hat{\mu} - \mu\| \|\hat{\pi} - \pi\|\right), \quad R_2 = O_{\mathbb{P}}\left(\frac{\sqrt{\sum_{g=1}^{G} \mathrm{Var}\left(\sum_{i=1}^{n_g} \hat{\varphi}(\mathbf{O}_{gi}) - \varphi(\mathbf{O}_{gi}) \mid D\right)}}{n}\right),$$

*where for a (potentially random) function* $f$ *of the observation,* $\|f\| = \sqrt{\int f^2(\boldsymbol{o}) d\mathbb{P}(\boldsymbol{o})}$. *Assuming* $R_1 + R_2 = o_{\mathbb{P}}\left(\sqrt{\Omega_n/n}\right)$, *we have*

$$\sqrt{\frac{n}{\Omega_n}} (\hat{\theta}_{DR} - \theta) \xrightarrow{d} N(0, 1).$$

By deriving a bound on the conditional variance term under the worst-case scenario of perfect within-cluster dependence, we have the following corollary.

**Corollary 1.** *Under the conditions in Theorem 1, the conditional variance can be bounded as*

$$\frac{\sqrt{\sum_{g=1}^{G} \text{Var}\left(\sum_{i=1}^{n_g} \hat{\varphi}(\mathbf{O}_{gi}) - \varphi(\mathbf{O}_{gi}) \mid D\right)}}{n} \leq \frac{\sqrt{\sum_{g=1}^{G} n_g^2 \|\hat{\varphi} - \varphi\|^2}}{n}.$$

*Consequently, under the following rate conditions*

$$\|\hat{\mu} - \mu\|\|\hat{\pi} - \pi\| = o_{\mathbb{P}}\left(\sqrt{\frac{\Omega_n}{n}}\right), \quad \|\hat{\varphi} - \varphi\| = o_{\mathbb{P}}\left(\sqrt{\frac{n\Omega_n}{\sum_{g=1}^{G} n_g^2}}\right),$$

*the asymptotic normality in Theorem 1 holds.*

In practice, the individual influence function $\varphi$ is often bounded, which implies $\mathbb{E}\left[|\varphi(\mathbf{O}_{gi})|^r\right] < \infty$. As we note above, Park and Kang (2021) also establishes the asymptotic normality of the DR estimator under clustered sampling, but require bounded cluster sizes $n_g \leq M < \infty$. We generalize the results in Park and Kang (2021) and allow each cluster size $n_g$ to diverge as $n \to \infty$, provided that (3) is satisfied. The second inequality in (3) is less restrictive for large $r$ since

$$\frac{\left(\sum_{g=1}^{G} n_g^r\right)^{2/r}}{n} \to \max_{g \leq G} \frac{n_g^2}{n}$$

as $r \to \infty$ and condition $\left(\sum_{g=1}^{G} n_g^r\right)^{2/r}/n \leq C$ is reduced to $\max_{g \leq G} n_g^2/n \leq C$, which is implied by the last inequality in (3). We also note that when (3) holds, the number of clusters $G \to \infty$ since

$$1 = \frac{\sum_{g=1}^{G} n_g}{n} \leq G \max_{1 \leq g \leq G} \frac{n_g}{n} \leq G \max_{1 \leq g \leq G} \frac{n_g^2}{n}.$$

In Condition 2 of Theorem 1, $\Omega_n$ is the asymptotic variance of $\sqrt{n}\hat{\theta}_{\text{DR}}$, which determines the final convergence rate. Condition 2 rules out degenerate cases where $\text{Var}(\sqrt{n}\hat{\theta}_{\text{DR}})$ vanishes. Condition 3 imposes requirements on the convergence rate of nuisance functions estimation. Different from the i.i.d. setting where the estimator achieves a $\sqrt{n}$-rate, $\Omega_n$ may diverge with $n$ when the within-cluster correlation is strong, resulting in a slower convergence rate. Consequently, compared with the rate condition $\|\hat{\mu} - \mu\|\|\hat{\pi} - \pi\| = o_{\mathbb{P}}(1/\sqrt{n})$ in the i.i.d. setting, the nuisance functions can be estimated at a slower rate in our clustered setting since the final target rate may also be slower.

In contrast to most existing work (Chen and Zhou, 2011; Yang, 2018; Alene et al., 2025), our results accommodate fully nonparametric and flexible modeling of the nuisance functions. In the literature, many results exist on regression function estimation for dependent observations, including GLM modeling (Daskalakis et al., 2019), wavelet-based methods (Yogendra P. Chaubey and Shirazi, 2013), kernel regression (Shimizu, 2024), random forests (Young and Bühlmann, 2025) and neural networks (Kohler and Krzyżak, 2023). The dependency structure among observations can be spatial, temporal, or induced by a social network (Kandiros et al., 2021). Notably, i.i.d.-based nonparametric and machine learning methods are commonly employed to study treatment effects in multilevel settings (Carvalho et al., 2019), despite the presence of within-cluster dependency, likely due to their simplicity (Park and Kang, 2021). While nonparametric machine learning methods in nuisance estimation help avoid model misspecification, their theoretical guarantees require further investigation depending on the specific dependence structure of the data.

As established in Theorem 1, the convergence rate of $\hat{\theta}_{\text{DR}}$ is $\sqrt{n/\Omega_n}$, which adapts to the degree of within-cluster dependence among the influence functions $\{\varphi(\mathbf{O}_{gi}) : 1 \leq i \leq n_g\}$. This behavior differs from most existing work on missing data or causal inference in clustered settings (e.g., Park and Kang, 2021), which often imposes bounded cluster sizes and can only achieve a $\sqrt{G}$-rate, corresponding to perfect within-cluster dependence. When within-cluster dependence is weak, $\hat{\theta}_{\text{DR}}$ can converge at a faster rate than $\sqrt{G}$. Additional examples in Appendix B further illustrate these conditions and convergence rates under various dependence structures.

To estimate the variance in this homogeneous sampling setting, denote $\tilde{\varphi}(\mathbf{O}_g) = \sum_{i=1}^{n_g} \varphi(\mathbf{O}_{gi})$. We can re-write $\Omega_n = \frac{1}{n} \sum_{g=1}^{G} \mathbb{E}[\tilde{\varphi}^2(\mathbf{O}_g)] - \frac{1}{n} \sum_{g=1}^{G} n_g^2 \theta^2$. A natural estimator for $\Omega_n$ is then given by

$$\hat{\Omega}_n = \frac{1}{n} \sum_{g=1}^{G} \left( \sum_{i=1}^{n_g} \hat{\varphi}(\mathbf{O}_{gi}) \right)^2 - \frac{1}{n} \sum_{g=1}^{G} n_g^2 \hat{\theta}_{DR}^2. \tag{4}$$

Under the conditions of Theorem 1, we can show that $\hat{\Omega}_n$ is a consistent estimator of $\Omega_n$ in the sense that $\hat{\Omega}_n / \Omega_n \xrightarrow{P} 1$. Therefore, $\hat{\Omega}_n/n$ can be used as a cluster-robust variance estimator for $\hat{\theta}_{\text{DR}}$ to perform statistical inference. A more robust—though computationally intensive—approach to variance estimation involves bootstrapping at the cluster level (Field and Welsh, 2007). Depending on the data-generating process, one may choose to resample both clusters and individuals (i.e., a two-stage bootstrap) or to resample clusters only. An alternative is the cluster wild bootstrap, which is well-suited for settings with heteroskedasticity, few clusters, or varying cluster sizes (MacKinnon and Webb, 2017). For a comprehensive discussion of resampling methods for clustered data, see Leeden et al. (2008).

## 4 Clustered Missing Data under Sequential Sampling

In this section, we relax the homogeneous sampling assumption and study the estimation problem in the presence of temporal dependency within each cluster. For example, in the context of LLM evaluation, each user may ask questions in a sequential manner. In this sequential setting, the missingness mechanism of the outcome $Y_{gt}$ at time $t$ may depend on the history (i.e., information before time $t$).

For cluster $g$ with cluster-level covariates $\mathbf{X}_g$, let $\overline{\mathbf{W}}_{gt} = (\mathbf{W}_{g1}, \ldots, \mathbf{W}_{gt}), \overline{\mathbf{R}}_{gt} = (R_{g1}, \ldots, R_{gt})$, $\overline{\mathbf{R}\mathbf{Y}}_{gt} = (R_{g1}Y_{g1}, \ldots, R_{gt}Y_{gt})$ denote the individual-level covariates, missing indicators and outcomes up to time $t$, respectively. The observations within the same cluster $\{\mathbf{O}_{g1}, \ldots, \mathbf{O}_{gn_g}\}$ are assumed to be generated sequentially. Let $\mathbf{H}_{gt} = (\overline{\mathbf{W}}_{gt}, \overline{\mathbf{R}}_{g,t-1}, \overline{\mathbf{R}\mathbf{Y}}_{g,t-1})$ denote the past history just prior to observing $R_{gt}, R_{gt}Y_{gt}$ at time $t$. The following sequential missing at random assumption is imposed on the data-generating process.

**Assumption 2** (Sequential missing at random). $R_{gt} \perp\!\!\!\perp Y_{gt} \mid \mathbf{X}_g, \mathbf{H}_{gt}$.

Assumption 2 implies that the missingness at time $t$ only depends on the history $\mathbf{H}_{gt}$ and cluster-level covariates $\mathbf{X}_g$. Denote $\pi_{gt}(\mathbf{X}_g, \mathbf{H}_{gt}) = \mathbb{P}(R_{gt} = 1 \mid \mathbf{X}_g, \mathbf{H}_{gt})$ and $\mu_{gt}(\mathbf{X}_g, \mathbf{H}_{gt}) = \mathbb{E}[R_{gt}Y_{gt} \mid \mathbf{X}_g, \mathbf{H}_{gt}, R_{gt} = 1]$. The data-generating process is as follows: First sample $G$ i.i.d. cluster-level covariates $\mathbf{X}_1, \ldots, \mathbf{X}_G \sim \mathbb{P}_{\mathbf{X}}$. For the $t$-th observation in the $g$-th cluster, we generate $\mathbf{W}_{gt}$ conditioned on the history up to time $t$: $\overline{\mathbf{W}}_{g,t-1}, \overline{\mathbf{R}}_{g,t-1}, \overline{\mathbf{R}\mathbf{Y}}_{g,t-1}$. The missing indicator $R_{gi}$ is then generated from Bernoulli$(\pi_{gt}(\mathbf{X}_g, \mathbf{H}_{gt}))$, following which $Y_{gt}$ is sampled from $\mathbb{P}_{Y_{gt}|\mathbf{X}_g, \mathbf{H}_{gt}, R_{gt}=1}$ with conditional mean $\mu_{gt}(\mathbf{X}_g, \mathbf{H}_{gt})$. The likelihood function of $\{\mathbf{O}_{gt} = (\mathbf{X}_g, \mathbf{W}_{gt}, R_{gt}, R_{gt}Y_{gt}), 1 \le t \le n_g, 1 \le g \le G\}$ is

$$\prod_{g=1}^{G} f_{\boldsymbol{x}}(\mathbf{X}_g) \prod_{t=1}^{n_g} \Big\{ f_{\boldsymbol{w}_{gt}}(\mathbf{W}_{gt} \mid \mathbf{X}_g, \mathbf{H}_{g,t-1}, R_{g,t-1}, R_{g,t-1}Y_{g,t-1})$$
$$\times \left[ \pi_{gt}(\mathbf{X}_g, \mathbf{H}_{gt}) f_{y_{gt}}(Y_{gt} \mid \mathbf{X}_g, \mathbf{H}_{gt}, R_{gt} = 1) \right]^{R_{gt}} (1 - \pi_{gt}(\mathbf{X}_g, \mathbf{H}_{gt}))^{1-R_{gt}} \Big\}.$$

Under Assumption 2, the average outcome that we are interested in is

$$\psi_n = \frac{1}{n} \sum_{g=1}^{G} \sum_{t=1}^{n_g} \mathbb{E}[Y_{gt}] = \frac{1}{n} \sum_{g=1}^{G} \sum_{t=1}^{n_g} \mathbb{E}[\mu_{gt}(\mathbf{X}_g, \mathbf{H}_{gt})].$$

Note that the variables $(\mathbf{X}_g, \mathbf{H}_{gt})$ no longer share the same distribution across different times and clusters. The regression function and propensity score are both cluster- and time-specific. Hence, several challenges arise in estimating $\psi_n$ by leveraging the nuisance functions:

1. Some clusters are small and do not support the modeling of cluster-level nuisance functions.

2. Different clusters have varying time steps depending on their size (e.g., some users may ask more questions than others). When only a small number of users ask more than $t$ questions (for large $t$), estimating the nuisance functions $\mu_{gt}$ and $\pi_{gt}$ becomes challenging.

3. The dimension of $\mathbf{H}_{gt}$ increases over time (i.e., the dimension of the arguments for these nuisance functions grows), which is important to note if one aims to simplify the modeling procedure by constructing unified models that are not cluster- or time-specific.

Given these challenges, we propose the following assumption to simplify estimation.

**Assumption 3.** *There exists an observed variable* $\mathbf{S}_{gt} \in \sigma(\mathbf{X}_g, \mathbf{H}_{gt})$ *and functions* $\pi, \mu : \mathcal{X} \times \mathcal{S} \to \mathbb{R}$ *such that*

$$\pi_{gt}(\mathbf{X}_g, \mathbf{H}_{gt}) = \pi(\mathbf{X}_g, \mathbf{S}_{gt}), \; \mu_{gt}(\mathbf{X}_g, \mathbf{H}_{gt}) = \mu(\mathbf{X}_g, \mathbf{S}_{gt}).$$

The variable $\mathbf{S}_{gt}$ can be viewed as a sufficient summary of the historical information up to time $t$. In practice, the choice of $\mathbf{S}_{gt}$ often requires domain knowledge. Common choices include average or cumulative measures of past information. For example, in mobile health studies, a user wears a fitness tracker that collects data daily. The device may fail to record data at time $t$ due to battery depletion, which depends on historical usage; in this case, $\mathbf{S}_{gt}$ could represent the cumulative device usage over the past few days. In educational testing or tutoring systems, whether a student attempts a question at time $t$ may depend on cumulative difficulty or frustration from earlier interactions. Here, $\mathbf{S}_{gt}$ might be defined as the number of incorrect attempts or a difficulty-adjusted score accumulated up to time $t$. In the LLM evaluation setting, whether users provide feedback may depend on their prior interactions with the system. Accordingly, $\mathbf{S}_{gt}$ can be constructed as the embedding of the concatenated conversation history $\overline{\mathbf{W}}_{gt}$ up to time $t$, or from the most recent $d$ conversations $\mathbf{W}_{g,t-d+1}, \ldots, \mathbf{W}_{gt}$. These embeddings remain of fixed length regardless of the length of the conversation history.

Assumption 3 also simplifies the data-generating process by assuming the missingness mechanism and the regression function of $Y_{gt}$ depend on the cluster-level covariates $\mathbf{X}_g$ and summarized information at time $t$, $\mathbf{S}_{gt}$, in the same way (i.e., they are not cluster- or time-specific). The doubly robust estimator of $\psi_n$ is then given by

$$\hat{\psi}_{\mathrm{DR}} = \frac{1}{n} \sum_{g=1}^{G} \sum_{t=1}^{n_g} \left[ \frac{R_{gt}(Y_{gt} - \hat{\mu}(\mathbf{X}_g, \mathbf{S}_{gt}))}{\hat{\pi}(\mathbf{X}_g, \mathbf{S}_{gt})} + \hat{\mu}(\mathbf{X}_g, \mathbf{S}_{gt}) \right], \tag{5}$$

where we slightly abuse the notation and still denote the influence function as $\varphi(\mathbf{Z}_{gi}) = \frac{R_{gt}(Y_{gt} - \mu(\mathbf{X}_g, \mathbf{S}_{gt}))}{\pi(\mathbf{X}_g, \mathbf{S}_{gt})} + \mu(\mathbf{X}_g, \mathbf{S}_{gt})$ with $\mathbf{Z}_{gi} = (\mathbf{X}_g, \mathbf{S}_{gi}, R_{gt}, R_{gt}Y_{gt})$ including both the observation and the summarized information $\mathbf{S}_{gt}$ at time $t$. While the idea of using summary statistics to simplify nuisance function modeling has been mentioned in Park and Kang (2021), our work formalizes this as an explicit assumption and establishes corresponding theoretical guarantees in the following theorem.

**Theorem 2.** *Under Assumption 2–3, further assume*

1. *For some* $r \geq 2$, $\{|\varphi(\mathbf{Z}_{gt})|^r, 1 \leq t \leq n_g, 1 \leq g \leq G\}$ *are uniformly integrable, i.e.,*

$$\lim_{M \to \infty} \sup_{g,t} \mathbb{E}\left[|\varphi(\mathbf{Z}_{gt})|^r I\left(|\varphi(\mathbf{Z}_{gt})| > M\right)\right] = 0.$$

*The cluster sizes and total sample size satisfy*

$$\frac{\left(\sum_{g=1}^{G} n_g^r\right)^{2/r}}{n} \leq C < \infty, \; \max_{g \leq G} \frac{n_g^2}{n} \to 0. \tag{6}$$

2. $\Omega_n = \frac{1}{n} \sum_{g=1}^{G} \mathrm{Var}\left(\sum_{t=1}^{n_g} \varphi(\mathbf{Z}_{gt})\right) \geq c > 0.$

3. $\hat{\pi}, \hat{\mu}$ *are estimated from a separate independent sample* $D$ *satisfying* $\hat{\pi} \geq \epsilon > 0$ .

*Then we have*

$$\hat{\psi}_{DR} - \psi_n = \frac{1}{n} \sum_{g=1}^{G} \sum_{t=1}^{n_g} (\varphi(\mathbf{Z}_{gt}) - \mathbb{E}[\varphi(\mathbf{Z}_{gt})]) + T_1 + T_2,$$

$$T_1 = O_{\mathbb{P}}\left(\frac{\sqrt{\sum_{g=1}^{G} \operatorname{Var}\left(\sum_{i=1}^{n_g} \hat{\varphi}(\mathbf{Z}_{gt}) - \varphi(\mathbf{Z}_{gt}) \mid D\right)}}{n}\right),$$

$$T_2 = O_{\mathbb{P}}\left(\frac{1}{n}\sum_{g=1}^{G}\sum_{t=1}^{n_g} \|\hat{\pi}(\mathbf{X}_g, \mathbf{S}_{gt}) - \pi(\mathbf{X}_g, \mathbf{S}_{gt})\|\|\hat{\mu}(\mathbf{X}_g, \mathbf{S}_{gt}) - \mu(\mathbf{X}_g, \mathbf{S}_{gt})\|\right).$$

*Assuming* $T_1 + T_2 = o_{\mathbb{P}}\left(\sqrt{\Omega_n/n}\right)$, *we have*

$$\sqrt{\frac{n}{\Omega_n}}(\hat{\psi}_{DR} - \psi_n) \xrightarrow{d} N(0,1).$$

Theorem 2 establishes the asymptotic normality of the doubly robust estimator when the observations $\mathbf{Z}_{gt}$ may follow heterogeneous distributions. In this setting, a uniform integrability condition—analogous to Lindeberg's condition when $r = 2$—is required to ensure no single term dominates the sum (Hansen and Lee, 2019). The conditional bias term $T_2$ depends on the average nuisance estimation error across all $n$ observations. We further provide a worst-case bound on both the empirical process and bias terms in the following corollary, similar to the result in Corollary 1.

**Corollary 2.** *Under the conditions in Theorem 2, we have the following bound on the error terms:*

$$\frac{\sqrt{\sum_{g=1}^{G} \operatorname{Var}\left(\sum_{i=1}^{n_g} \hat{\varphi}(\mathbf{Z}_{gt}) - \varphi(\mathbf{Z}_{gt}) \mid D\right)}}{n} = O_{\mathbb{P}}\left(\frac{1}{n}\sqrt{\sum_{g=1}^{G} n_g^2 \sup_{\mathbf{z}} \mathbb{E}_D\left[(\hat{\varphi}(\mathbf{z}) - \varphi(\mathbf{z}))^2\right]}\right),$$

$$\frac{1}{n}\sum_{g=1}^{G}\sum_{t=1}^{n_g} \|\hat{\pi}(\mathbf{X}_g, \mathbf{S}_{gt}) - \pi(\mathbf{X}_g, \mathbf{S}_{gt})\|\|\hat{\mu}(\mathbf{X}_g, \mathbf{S}_{gt}) - \mu(\mathbf{X}_g, \mathbf{S}_{gt})\|$$

$$= O_{\mathbb{P}}\left(\sqrt{\sup_{\boldsymbol{x},\boldsymbol{s}} \mathbb{E}_D\left[(\hat{\pi}(\boldsymbol{x},\boldsymbol{s}) - \pi(\boldsymbol{x},\boldsymbol{s}))^2\right] \sup_{\boldsymbol{x},\boldsymbol{s}}\left[\mathbb{E}_D(\hat{\mu}(\boldsymbol{x},\boldsymbol{s}) - \mu(\boldsymbol{x},\boldsymbol{s}))^2\right]}\right),$$

*where the supremum is taken over the support of the corresponding variables. Consequently, if we further assume*

$$\sqrt{\sup_{\mathbf{z}} \mathbb{E}_D\left[(\hat{\varphi}(\mathbf{z}) - \varphi(\mathbf{z}))^2\right]} = o\left(\sqrt{\frac{n\Omega_n}{\sum_{g=1}^{G} n_g^2}}\right),$$

$$\sqrt{\sup_{\boldsymbol{x},\boldsymbol{s}} \mathbb{E}_D[(\hat{\mu}(\boldsymbol{x},\boldsymbol{s}) - \mu(\boldsymbol{x},\boldsymbol{s}))^2] \sup_{\boldsymbol{x},\boldsymbol{s}} \mathbb{E}_D[(\hat{\pi}(\boldsymbol{x},\boldsymbol{s}) - \pi(\boldsymbol{x},\boldsymbol{s}))^2]} = o\left(\sqrt{\frac{\Omega_n}{n}}\right),$$

*the asymptotic normality in Theorem 2 holds.*

Finally, our approach of using a summary to simplify the modeling of within-cluster dependence can be extended to other settings. For instance, when clusters are defined by a network model, characteristics of an individual's neighbors (e.g., degree or sum of edge weights) can be instrumental and serve as the summary information in modeling the missingness mechanism. Similar asymptotic normality results can be derived by leveraging the central limit theorem for clustered data.

## 5  Simulation Study

This section presents simulation studies to illustrate our theoretical results; full details are provided in Appendix C. In the first study, we compare confidence intervals using i.i.d.-based and cluster-robust variance estimators. Figure 1(a) shows that only the cluster-robust approach achieves nominal 95% coverage, underscoring the importance of accounting for cluster dependence. In the second study, we evaluate the impact of incorporating historical information when modeling missingness in sequential data. As shown in Figure 1(b), modeling the missingness mechanism with relevant historical summaries improves performance compared to using only current information or ignoring missingness, highlighting the importance of leveraging past interactions in sequential settings.

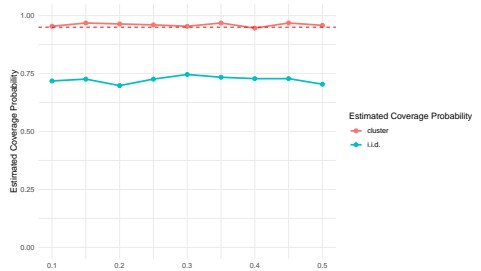

(a) Estimated coverage probability of CIs based on different variance estimators.

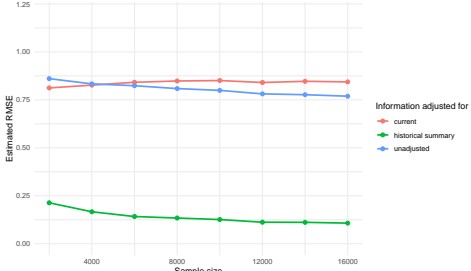

(b) Estimated RMSE of estimators adjusting for different information.

Figure 1: Simulation results.

# 6 Real Data Analysis

The alignment of AI systems with human preferences is a critical area of research. A key challenge for prominent methods such as Preference Flow Matching (Kim et al., 2025) and Reinforcement Learning from Human Feedback (RLHF) (Bai et al., 2022; Casper et al., 2023) is that human annotations for evaluating alignment are often costly and incomplete. In this context, we illustrate our methods using the OpenAssistant Conversations dataset (Köpf et al., 2023), a human-annotated conversation corpus structured as trees, with each tree representing a conversation cluster and its messages as individual observations. The cleaned dataset includes 9,808 trees and 81,937 messages, with message-level covariates such as `content` and `role`, and tree-level covariates such as `language`. We focus on annotations for `quality`, `creativity`, `humor`, and `toxicity`, and consider two missingness mechanisms: (i) missingness depends only on observed covariates (Assumption 1), or (ii) missingness depends on the conversation history (Assumption 2) leading up to the message node, i.e., the path from the root to the message within the conversation tree. The individual-level covariates $\mathbf{W}_{gi}$ are `message embeddings` and `role`, and the cluster-level covariate $\mathbf{X}_g$ is `language`; missingness is simulated via a logistic model. We estimate average annotation scores using three methods: (1) naive i.i.d.-based confidence intervals only using data with annotations observed, (2) doubly robust estimation assuming independence, and (3) doubly robust estimation with cluster-robust variance estimation as in Eq. (4). More details about the dataset, missingness and estimation can be found in Appendix D. Figure 2 shows the resulting confidence intervals for each annotation type, compared to the ground truth average $\frac{1}{n} \sum_{g=1}^{G} \sum_{i=1}^{n_g} Y_{gi}$.

Figure 2 shows that unadjusted estimates, which ignore observations with missing annotations, are biased and yield confidence intervals that fail to cover the true average. This bias arises because covariates $\mathbf{W}_{gi}$ and $\mathbf{X}_g$ influence both the missingness and the outcome, acting as confounders. The doubly robust adjusted estimates correct for this bias and are closer to the ground truth. Among the adjusted methods, confidence intervals assuming independence are narrower but may undercover due to within-cluster dependence. For example, in Figure 2(c), the interval for `humor` under the i.i.d. assumption fails to cover the true value, while the cluster-robust interval does. These results underscore the need to account for cluster dependence when constructing valid confidence intervals.

# 7 Discussion

This paper studies mean estimation with missing responses under cluster dependence, focusing on the widely used doubly robust estimator and establishing its theoretical guarantees in clustered settings. We mainly consider two primary scenarios—homogeneous sampling and sequential dependence—but our methods extend to more general structures like network dependence. Our theoretical and empirical results highlight the importance of properly accounting for cluster dependence, with valuable implications for applications such as LLM evaluation using limited human annotations.

There are several directions for future research. First, the doubly robust estimator can be unstable when propensity scores are near zero; using balancing weights may offer a more stable alternative (Ben-Michael et al., 2024), and its performance under cluster or sequential dependence requires further study. Second, estimating means in target populations with only covariate information—such

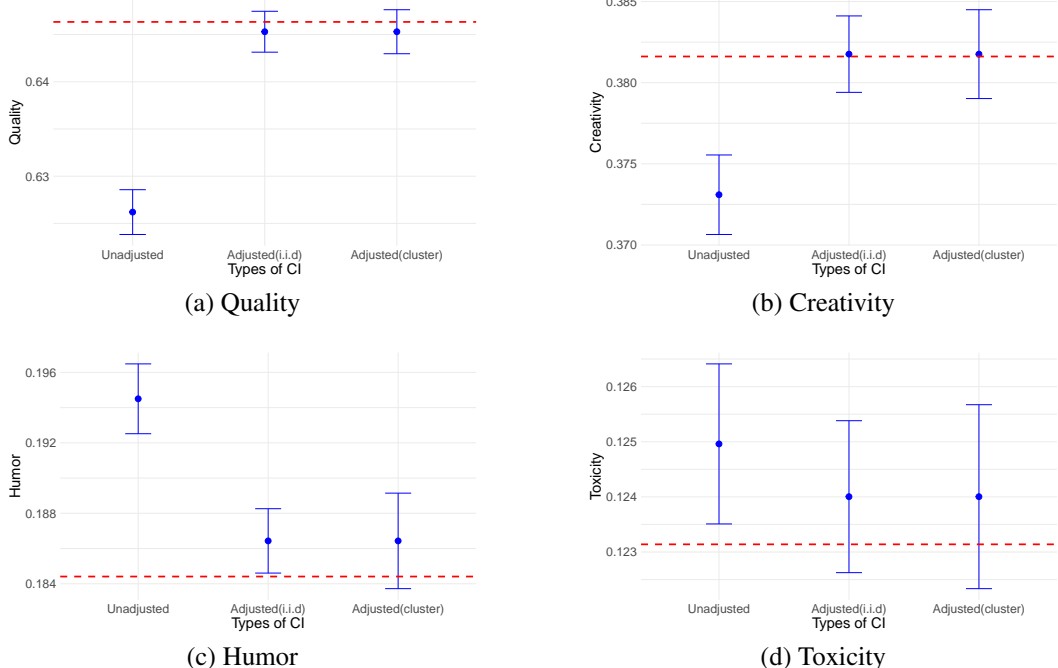

Figure 2: Confidence intervals for average human annotations on quality, creativity, humor, and toxicity under homogeneous sampling. The red dashed line is the ground truth average.

as evaluating a different language model without human annotations—is related to covariate shift (Sugiyama and Kawanabe, 2012) and generalization/transportation (Dahabreh et al., 2020; Zeng et al., 2023). Investigating these extensions is also a promising direction for future work. Another interesting and important direction is to develop theory and methods for flexible regression and propensity score estimation under clustered settings, to enable more reliable ATE estimation using nonparametric doubly robust methods.

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

# A  Related Work

In the i.i.d. setting, it is well-established that estimating the average treatment effect under the exchangeability assumption is equivalent to estimating the mean in a missing data problem under missing at random (MAR), with the treatment assignment analogous to a missingness indicator (Tsiatis, 2006). However, when cluster dependence is present, the problem setup, estimands of interest, and analytical techniques diverge substantially.

In the literature on cluster-randomized trials (Balzer et al., 2019; Wang et al., 2021; Schochet, 2022), treatments are assigned at the cluster level, so all individuals within a cluster receive the same treatment. In contrast, our setting involves individual-level missingness indicators within clusters, allowing each individual to have their own observed or missing outcome.

Other works in causal inference consider individual-level treatments within clusters but focus on interference—where one individual's outcome depends on the treatment assignments of others in the same cluster (Liu et al., 2019; Vazquez-Bare, 2023). Under interference, even a single unit's treatment status can affect the observability of potential outcomes for the entire cluster, which differs fundamentally from our MAR-based framework. In our setting, missingness is modeled directly via $R_{gi}$, and each individual's outcome may be observed or unobserved regardless of others in the same cluster.

While some doubly robust methods for clustered data exist in the causal inference and missing data literature (Chen and Zhou, 2011; Yang, 2018; Alene et al., 2025), these approaches often rely on parametric assumptions for nuisance functions $\pi$ and $\mu$. In contrast, our approach is fully nonparametric and accommodates modern machine learning techniques for nuisance estimation.

Finally, most existing analyses do not leverage recent central limit theorems (CLTs) for clustered data (Hansen and Lee, 2019), which limits their applicability to settings with equal or bounded cluster sizes and often leads to overly conservative $\sqrt{G}$-rate results. For instance, in the Appendix of Park and Kang (2021); Alene et al. (2025), theoretical guarantees for the doubly robust estimator in multilevel observational studies are established, but they require bounded cluster sizes and only achieve a $\sqrt{G}$-rate. In contrast, our analysis accommodates diverging cluster sizes and achieves a convergence rate of $\sqrt{n/\Omega_n}$, which can be faster than the $\sqrt{G}$-rate when within-cluster dependence is weak.

# B  Illustrative Examples of Convergence Rates

Consider a setting where all clusters are of equal size $n_g = n^\alpha$ for some $\alpha \in (0,1)$, and the total number of clusters is $G = n^{1-\alpha}$. Under this setup, condition (3) simplifies to $\alpha \leq (r-2)/(2r-2)$.

We first provide two examples under the homogeneous sampling setting as in Section 3.

**Example 1** (i.i.d. sampling). *Consider a special case where individual influence functions $\{\varphi(\mathbf{O}_{gi}), 1 \leq i \leq n_g\}$ are independent within clusters This scenario could arise when $\pi$ and $\mu$ are functions only of $\mathbf{W}$ and $\{\mathbf{W}_{gi}, 1 \leq i \leq n_g\}$ are independent. In this case, we have*

$$\Omega_n = \mathrm{Var}(\varphi(\mathbf{O}_{gi})),$$

*which is a constant and we assume it is positive. The conditions on nuisance estimation to achieve $\sqrt{n}$-rate in Theorem 1 are*

$$\|\hat{\mu} - \mu\|\|\hat{\pi} - \pi\| = o_{\mathbb{P}}\left(1/\sqrt{n}\right), \ \|\hat{\varphi} - \varphi\| = o_{\mathbb{P}}\left(1\right),$$

*which are the same as conditions for the doubly robust estimator to be $\sqrt{n}$-consistent in the i.i.d. setting. However, Corollary 1 requires the stronger conditions:*

$$\|\hat{\mu} - \mu\|\|\hat{\pi} - \pi\| = o_{\mathbb{P}}\left(1/\sqrt{n}\right), \ \|\hat{\varphi} - \varphi\| = o_{\mathbb{P}}\left(1/\sqrt{n^\alpha}\right).$$

*The need for these stronger conditions in Corollary 1 arises from our worst-case analysis of the variance when bounding the empirical process term, which may not be tight when independence also holds within clusters.*

**Example 2** (Perfect correlation within cluster). *Consider another special case where, for each cluster $g$, the individual influence functions and their estimates are all equal (i.e., $\varphi(\mathbf{O}_{g1}) = \cdots = \varphi(\mathbf{O}_{gn_g})$*

*and $\hat{\varphi}(\mathbf{O}_{g1}) = \cdots = \hat{\varphi}(\mathbf{O}_{gn_g}))$, so within-cluster dependency is perfect. We have*

$$\Omega_n = n^\alpha \operatorname{Var}(\varphi(\mathbf{O}_{gi})),$$

*and the convergence rate is $\sqrt{\Omega_n/n} \asymp n^{-(1-\alpha)/2} = G^{-1/2}$. Intuitively, the effective sample size is $G$ since we effectively only have repeated measures within each cluster. The conditions on nuisance estimation required to achieve the $\sqrt{G}$-rate in Theorem 1 are*

$$\|\hat{\mu} - \mu\|\|\hat{\pi} - \pi\| = o_{\mathbb{P}}\left(\frac{1}{\sqrt{G}}\right), \quad \|\hat{\varphi} - \varphi\| = o_{\mathbb{P}}(1).$$

We provide another example where the influence functions of different observations have weak stationary dependence in the temporal setting (Section 4).

**Example 3** (Weak stationary dependence). *Consider the case where within each cluster, the sequence of individual influence functions satisfies the following weak stationary condition:*

$$\operatorname{Var}(\varphi(\mathbf{Z}_{gt})) = 1, \ \operatorname{Cov}(\varphi(\mathbf{Z}_{gt}), \varphi(\mathbf{Z}_{gs})) = 1/|t-s|, \ 1 \le s,t \le n_g, s \neq t.$$

*Recall that in all examples we assume $n_g = n^\alpha, G = n^{1-\alpha}$. Simple calculations yield*

$$\Omega_n = \frac{1}{n}G\left[2n^\alpha \sum_{t=1}^{n^\alpha-1} \frac{1}{t} - n^\alpha + 2\right] \asymp \log n$$

*and the convergence rate of $\hat{\psi}$ is $\sqrt{\log n/n}$. The rate condition on nuisance function estimation is*

$$\sqrt{\sup_{\mathbf{z}} \mathbb{E}_D\left[(\hat{\varphi}(\mathbf{z}) - \varphi(\mathbf{z}))^2\right]} = o\left(\sqrt{\frac{\log n}{n^\alpha}}\right),$$

$$\sqrt{\sup_{\boldsymbol{x},\boldsymbol{s}} \mathbb{E}_D[(\hat{\mu}(\boldsymbol{x},\boldsymbol{s}) - \mu(\boldsymbol{x},\boldsymbol{s}))^2] \sup_{\boldsymbol{x},\boldsymbol{s}} \mathbb{E}_D[(\hat{\pi}(\boldsymbol{x},\boldsymbol{s}) - \pi(\boldsymbol{x},\boldsymbol{s}))^2]} = o\left(\sqrt{\frac{\log n}{n}}\right).$$

Finally, we provide an example that illustrates the impact of heterogeneous cluster sizes.

**Example 4** (Heterogeneous cluster sizes). *Consider a setting with two types of cluster sizes: size $1$ and size $n^\alpha$. There are $n/2$ clusters of the first type and $n^{1-\alpha}/2$ clusters of the second type, so the total number of clusters is*

$$G = \frac{n}{2} + \frac{n^{1-\alpha}}{2} \asymp n.$$

*Within each cluster, assume the individual influence functions and their estimates are all identical with unit variance. Then*

$$\Omega_n = \frac{n + n^{2\alpha} \cdot n^{1-\alpha}}{2n} \asymp n^\alpha,$$

*and the resulting convergence rate is*

$$\sqrt{\frac{\Omega_n}{n}} \asymp n^{-(1-\alpha)/2},$$

*which is slower than both $\sqrt{n}$ and $\sqrt{G}$, since $G \asymp n$. The corresponding rate condition for nuisance estimation is*

$$\|\hat{\mu} - \mu\|\|\hat{\pi} - \pi\| = o_{\mathbb{P}}\left(n^{-(1-\alpha)/2}\right), \quad \|\hat{\varphi} - \varphi\| = o_{\mathbb{P}}(1).$$

*This example highlights the importance of accounting for heterogeneous cluster sizes. Although the total number of clusters $G$ is large and of the same order as $n$, the convergence rate is driven by the relatively small number of large clusters, within which the correlation is perfect.*

## C   Simulation Details

In this section, we provide details for simulation studies that illustrate our theoretical results. In Appendix C.1, we provide details of numerical experiments that highlight the importance of accounting for cluster dependence and using a cluster-robust variance estimator to ensure proper coverage probabilities of confidence intervals. Additionally, in Appendix C.2, we present details of numerical experiments demonstrating the critical role of historical information in adjusting for missingness in a sequential setting.

## C.1 Homogeneous Sampling

Consider the following data-generating process: For each cluster $g$, the cluster-level covariate $X_g \sim N(0,1)$. Then the individual-level covariates $\mathbf{W}_g \sim N(\mathbf{1}_{n_g} X_g, \sigma^2 \Sigma)$ given $X_g$, where $\Sigma_{ij} = \rho^{|i-j|}$ for $\rho = 0.8, \sigma^2 = 4$. For each individual $i$, the missing indicator $R_{gi}$ is sampled from a Bernoulli distribution with mean $\pi(X_g, W_{gi}) = \text{logistic}(X_g + 0.5 W_{gi})$ and the outcome $Y_{gi}$ is sampled from $N(-X_g + W_{gi} + 0.5, 1)$. The average outcome is $\theta = 0.5$. In this experiment, we evaluate the necessity of considering the cluster structure in the estimation by comparing the coverage probability of confidence intervals based on two different variance estimators. The first variance estimator is $\hat{\sigma}_1^2 = \frac{1}{n-1} \sum_{g=1}^{G} \sum_{i=1}^{n_g} \left( \hat{\varphi}(\mathbf{O}_{gi}) - \hat{\theta}_{DR} \right)^2$, which is a consistent estimator of variance if observations $\{\mathbf{O}_{gi}, 1 \leq i \leq n_g, 1 \leq g \leq G\}$ are independent. The second estimator is $\hat{\sigma}_2^2 = \hat{\Omega}/n$ with $\hat{\Omega}$ given by (4) and takes the cluster structure into account. In each replication of experiment, we generate the data with total sample size $n = 10000$ and cluster size $n_g = n^\alpha$ for $\alpha \in \{0.1, 0.15, \ldots, 0.5\}$, compute the doubly robust estimator $\hat{\theta}_{DR}$ and construct Wald-confidence intervals based on $\hat{\sigma}_1^2, \hat{\sigma}_2^2$. We then repeat the process $M = 500$ times and estimate the coverage probability of the $95\%$ confidence intervals obtained. The results are summarized in Figure 1(a).

Figure 1(a) shows that the confidence intervals based on $\hat{\sigma}_2^2$ attain the nominal coverage probability of 0.95, as they appropriately account for the cluster dependence in the data. In contrast, the confidence intervals based on $\hat{\sigma}_1^2$ suffer from lower coverage probabilities than the nominal level, because $\hat{\sigma}_1^2$ ignores the cluster structure and consequently underestimates the variance.

## C.2 Sequential Sampling

Consider the following data-generating process: For each cluster $g$, the cluster-level covariate $X_g \sim N(0,1)$. The individual-level covariates are generated sequentially from an AR(2) process. Specifically,

$$\mathbf{W}_{gt} = \mathbf{A}_1 \mathbf{W}_{g,t-1} + \mathbf{A}_2 \mathbf{W}_{g,t-2} + \boldsymbol{\epsilon}_t, \; \boldsymbol{\epsilon}_t \sim N(0, 4\mathbf{I}_2).$$

Let $\mathbf{S}_{gt} = \left( \max_{1 \leq s \leq t, 1 \leq k \leq 2} W_{gsk}, \min_{1 \leq s \leq t, 1 \leq k \leq 2} W_{gsk}, \frac{1}{t} \sum_{s=1}^{t} \mathbf{W}_{gs} \right) \in \mathbb{R}^4$ be the summary of past information up to time $t$. For each time $t$, the missing indicator $R_{gt}$ is sampled from a Bernoulli distribution with mean $\pi(X_g, \mathbf{S}_{gt}) = \text{logistic}(X_g + (1, 0.8, -0.5, 0.3)^\top \mathbf{S}_{gt})$ and the outcome $Y_{gi}$ is sampled from $N(-X_g + (1, 1, -0.5, -0.4)^\top \mathbf{S}_{gt} + 1, 1)$. The average outcome is $\psi = 1$. In this experiment, we demonstrate the importance of adjusting for a useful summary of past information in modeling the missingness mechanism by comparing two estimators. The first one models the missingness mechanism $\pi$ as a function of $X_g, \mathbf{W}_{gt}$ while the second fits $\pi$ as a function of $X_g, \mathbf{S}_{gt}$. We also include the unadjusted estimator as a baseline. In each replication of the experiment, we generate the data with total sample size $n \in \{2000, 4000, \ldots, 16000\}$ and cluster size $n_g = n^{0.4}$, compute two doubly robust estimators $\hat{\psi}_{DR}$ adjusting for different information and evaluate the estimation error. We then repeat the process $M = 500$ times and estimate the Rooted-Mean-Squared-Error (RMSE) as

$$\hat{\text{RMSE}} = \sqrt{\frac{1}{M} \sum_{m=1}^{M} (\hat{\psi}_{DR}^m - \psi)^2}.$$

The results are summarized in Figure 1(b). As shown in Figure 1(b), the estimator that models the missingness mechanism using the correct historical information outperforms the one that adjusts only for the current information at each time $t$. This highlights the importance of incorporating relevant past information when modeling missingness in a sequential setting, such as users' sequential interactions with the system.

## D  Details for the Real Data

In this section, we provide more discussion on the background, implementation details and additional results of the real data analysis.

The alignment of AI systems with human values, intentions, and preferences is a crucial area of AI research. Techniques such as Preference Flow Matching (Kim et al., 2025) and Reinforcement Learning

from Human Feedback (RLHF) (Bai et al., 2022; Casper et al., 2023) have been developed to enhance the performance of LLMs across various applications. However, before focusing on improvement strategies, the first step is to assess how well an AI system aligns with human preferences based on available annotations. Human annotations serve as valuable tools for evaluating AI performance, yet they are often expensive and difficult to collect at scale. The purpose of Section 6 is to illustrate our methods using a conversational dataset where human annotations are missing for some observations.

Our analysis focuses on the OpenAssistant Conversations dataset (Köpf et al., 2023), a publicly available human-generated and human-annotated assistant-style conversation corpus [1]. The dataset is structured as conversation trees, where each tree begins with an initial prompt message (root node) that can have multiple child messages as replies, which in turn can have their own responses. Due to this hierarchical structure, messages within the same conversation tree are highly correlated, and we model each conversation tree as a cluster, containing many messages as individuals within the cluster.

The cleaned dataset consists of 9,808 conversation trees with a total of 81,937 messages. Message-level covariates include the content and `role` of the message, which indicates whether a message was generated by the prompter or the assistant, while conversation-level covariates include `language`, with English and Spanish being the most frequently observed languages. Each message is also annotated with multiple labels assessing different aspects, which serve as evaluation scores. In our analysis, we focus on annotations for quality, creativity, humor, and toxicity.

We first illustrate our methods in Section 3, where the missingness of annotations for each message depends only on its own content and the characteristics of the conversation tree to which it belongs directly (Assumption 1). Let the individual-level covariate $\mathbf{W}_{gi}$ represent the embedding of the $i$-th message in the $g$-th conversation tree along with its `role`. In this work, we use `BAAI/bge-small-en-v1.5` embeddings from Hugging Face $\mathbf{W}_{gi}$, which are fine-tuned specifically for embedding tasks. Additionally, we incorporate the `language` of each conversation tree as the cluster-level covariate $\mathbf{X}_g$. The missingness indicator for human annotations, $R_{gi}$, is then simulated from a logistic model satisfying $\mathbb{E}[R_{gi} \mid \mathbf{W}_{gi}, \mathbf{X}_g] = \text{expit}(\mathbf{W}_{gi}^{\top}\boldsymbol{\beta})$, with $\boldsymbol{\beta}$ being a randomly generated coefficient vector. This process mimics how human reviewers may decide whether to provide annotations based on message content and contextual factors. Let $Y_{gi}$ represent the annotations, which are treated as missing when $R_{gi} = 0$. We construct three types of confidence intervals for the average human annotations in our results. The first method restricts the analysis to messages with $R_{gi} = 1$ (i.e., ignoring messages with missingness) and applies the CLT for i.i.d. data. The second method applies the doubly robust estimator (2) to adjust for missingness but estimates variance under the assumption of independent observations. The third method further adopts a cluster-robust variance estimation approach as in (4). Sample splitting is used, and all nuisance functions are estimated from half of the sample by the SuperLearner (Polley et al., 2024) incorporating a generalized linear model and random forest. We plot these confidence intervals for annotations on quality, creativity, humor, and toxicity in Figure 2, with the dashed horizontal line $\frac{1}{n}\sum_{g,i} Y_{gi}$ serving as the ground truth.

We further illustrate the use of summary statistics from conversation history to adjust for missingness in Section 4. For each message, we assume that the probability of missing annotations depends on the conversation history up to that node in the conversation tree (i.e., the path from the root node to the message node). Let $\mathbf{S}_{gt}$ represent the embedding of the conversation history, aggregating conversations from all ancestor messages of the $t$-th message in the $g$-th conversation tree, along with its `role`. The cluster-level covariate is `language`. The missingness indicator is simulated using a logistic model $\mathbb{E}[R_{gt} \mid \mathbf{S}_{gt}, \mathbf{X}_g] = \text{expit}(\mathbf{S}_{gt}^{\top}\boldsymbol{\beta})$. This setup mimics how human reviewers may decide whether to provide annotations based on prior message content and contextual factors. We construct three types of confidence intervals for the average human annotation scores. The first method restricts the analysis to messages with $R_{gt} = 1$ (i.e., ignoring messages with missingness) and applies the CLT under an i.i.d. assumption. The second method applies the doubly robust estimator (5) to adjust for missingness but estimates variance under the assumption that the influence functions $\varphi(\mathbf{Z}_{gt})$ are independent. However, this confidence interval is not valid, as it ignores within-cluster dependence; we include it only for reference. The third method adopts a bootstrap-based variance estimation approach that accounts for the cluster structure. The bootstrap procedure takes approximately 20 hours per outcome on a 12-core CPU machine. We plot these confidence intervals for annotations on quality, creativity, humor, and toxicity in Figure 3, with the dashed horizontal line $\frac{1}{n}\sum_{g,t} Y_{gt}$ serving as the ground truth.

---

[1] Available at https://huggingface.co/datasets/OpenAssistant/oasst1

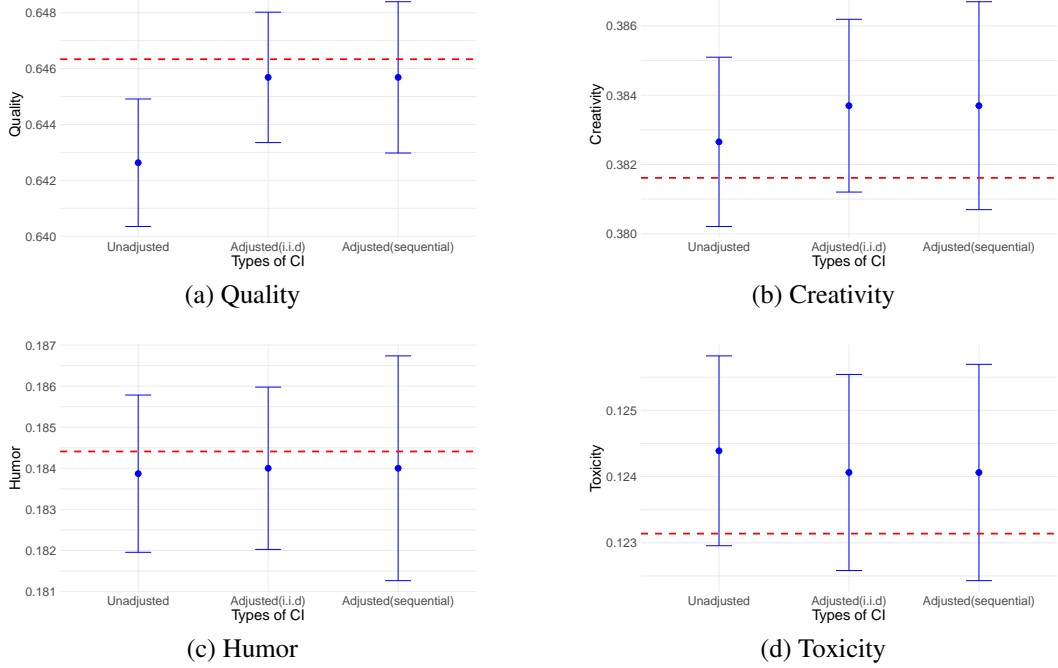

$$\text{(a) Quality} \qquad\qquad \text{(b) Creativity}$$

$$\text{(c) Humor} \qquad\qquad \text{(d) Toxicity}$$

Figure 3: Confidence intervals for average human annotations on quality, creativity, humor, and toxicity under the sequential structure. The red dashed line is the ground truth average.

As shown in Figure 3, the estimates that adjust for missingness are closer to the true sample average, represented by the red dashed line, for quality, humor, and toxicity, compared to the unadjusted estimates. This suggests that accounting for missingness effectively reduces estimation bias. Additionally, the confidence intervals that account for potential cluster dependence within conversation trees are at least 10% wider than those constructed under the i.i.d. assumption. This indicates that variance estimators based on the i.i.d. assumption underestimate the variation, highlighting the importance of using a cluster-robust approach for variance estimation.

## E   Additional Simulation Results

In this section, we present additional simulation results for the plug-in (regression) estimator

$$\hat{\theta}_{\text{OR}} = \frac{1}{n} \sum_{g=1}^{G} \sum_{i=1}^{n_g} \hat{\mu}(\mathbf{X}_g, \mathbf{W}_{gi}),$$

the IPW estimator

$$\hat{\theta}_{\text{IPW}} = \frac{1}{n} \sum_{g=1}^{G} \sum_{i=1}^{n_g} \frac{R_{gi} Y_{gi}}{\hat{\pi}(\mathbf{X}_g, \mathbf{W}_{gi})},$$

and the doubly robust estimator in equation (2). Our focus is on evaluating their performance under model misspecification in the presence of clustered data.

The data-generating process follows the homogeneous sampling setup described in Appendix C, with the regression function and propensity score given by

$$\mu(X_g, W_{gi}) = -X_g + W_{gi}^2, \quad \pi(X_g, W_{gi}) = \text{logistic}(X_g + 0.5 W_{gi}^2).$$

The true average outcome is $\theta = 5$. To assess estimator performance under misspecification, we consider scenarios in which $\mu$ and/or $\pi$ are misspecified by modeling the quadratic term in $W_{gi}$ as linear. We generate samples of size $n \in \{1000, 10000\}$ with varying cluster sizes, apply each estimator under different model specifications, and compute the mean squared error (MSE). The results are summarized in Tables 1–3.

| | Regression estimator | IPW estimator | DR estimator |
|---|---|---|---|
| $\mu$ correct, $\pi$ correct | 0.0562 | 0.0561 | 0.0563 |
| $\mu$ correct, $\pi$ wrong | 0.0563 | 2.7023 | 0.0563 |
| $\mu$ wrong, $\pi$ correct | 2.0356 | 0.0563 | 0.0572 |
| $\mu$ wrong, $\pi$ wrong | 2.0603 | 2.6896 | 2.5765 |

Table 1: Mean squared error (MSE) of regression, IPW, and DR Estimators under potential nuisance misspecification with sample size $n = 10000, n_g = 100$.

| | Regression estimator | IPW estimator | DR estimator |
|---|---|---|---|
| $\mu$ correct, $\pi$ correct | 0.0230 | 0.0231 | 0.0230 |
| $\mu$ correct, $\pi$ wrong | 0.0230 | 2.3882 | 0.0230 |
| $\mu$ wrong, $\pi$ correct | 2.1082 | 0.0232 | 0.0233 |
| $\mu$ wrong, $\pi$ wrong | 2.1075 | 2.3914 | 2.3102 |

Table 2: Mean squared error (MSE) of regression, IPW, and DR Estimators under potential nuisance misspecification with sample size $n = 10000, n_g = 10$.

The conclusions in this clustered setting are similar to those in the classical i.i.d. setting. When the outcome model $\mu$ is misspecified, the plug-in (regression) estimator $\hat{\theta}_{\mathrm{OR}}$ becomes inconsistent. Similarly, the consistency of the IPW estimator $\hat{\theta}_{\mathrm{IPW}}$ relies on correct specification of the propensity score $\pi$. In contrast, the doubly robust estimator $\hat{\theta}_{\mathrm{DR}}$, which models both the outcome and the missingness mechanism, remains consistent as long as either $\mu$ or $\pi$ is correctly specified. This aligns with the theoretical guarantees established in Theorem 1.

## F   Proof of Theorem 1

Since the observations $\{\mathbf{O}_{gi}, 1 \leq i \leq n_g, 1 \leq g \leq G\}$ share the same distribution, we have the following decomposition of error

$$
\begin{aligned}
\hat{\theta}_{DR} - \theta &= \frac{1}{n} \sum_{g=1}^{G} \sum_{i=1}^{n_g} (\varphi(\mathbf{O}_{gi}) - \mathbb{E}[\varphi(\mathbf{O}_{gi})]) \\
&+ \frac{1}{n} \sum_{g=1}^{G} \sum_{i=1}^{n_g} (\hat{\varphi}(\mathbf{O}_{gi}) - \varphi(\mathbf{O}_{gi}) - \mathbb{P}[\hat{\varphi}(\mathbf{O}_{gi}) - \varphi(\mathbf{O}_{gi})]) \\
&+ \mathbb{P}[\hat{\varphi}(\mathbf{O}_{gi}) - \varphi(\mathbf{O}_{gi})],
\end{aligned}
$$

where for a potentially random function $f$ of $\mathbf{O}$, $\mathbb{P}[f(\mathbf{O})] = \int f(o)d\mathbb{P}(o)$ so only the randomness of $\mathbf{O}$ is averaged over. For the first CLT term, by the central limit theorem for clustered data (Hansen and Lee, 2019)[Theorem 2], under the assumptions in our Theorem 1 we have

$$
\sqrt{\frac{n}{\Omega_n}} \frac{1}{n} \sum_{g=1}^{G} \sum_{i=1}^{n_g} (\varphi(\mathbf{O}_{gi}) - \mathbb{E}[\varphi(\mathbf{O}_{gi})]) \xrightarrow{d} N(0,1),
$$

and thus the order is

$$
\frac{1}{n} \sum_{g=1}^{G} \sum_{i=1}^{n_g} (\varphi(\mathbf{O}_{gi}) - \mathbb{E}[\varphi(\mathbf{O}_{gi})]) = O_{\mathbb{P}} \left( \sqrt{\frac{\Omega_n}{n}} \right).
$$

|  | Regression estimator | IPW estimator | DR estimator |
|---|---|---|---|
| $\mu$ correct, $\pi$ correct | 0.3247 | 0.3271 | 0.3248 |
| $\mu$ correct, $\pi$ wrong | 0.3232 | 5.3693 | 0.3246 |
| $\mu$ wrong, $\pi$ correct | 1.8736 | 0.3267 | 0.3470 |
| $\mu$ wrong, $\pi$ wrong | 1.9578 | 5.6812 | 5.3977 |

Table 3: Mean squared error (MSE) of regression, IPW, and DR Estimators under potential nuisance misspecification with sample size $n = 1000, n_g = 31$.

For the second empirical process term, by Markov's inequality, we have (conditioning on $D$ that is used to estimate the nuisance functions)

$$\mathbb{P}\left( \frac{1}{n} \sum_{g=1}^{G} \sum_{i=1}^{n_g} (\hat{\varphi}(\mathbf{O}_{gi}) - \varphi(\mathbf{O}_{gi}) - \mathbb{P}[\hat{\varphi}(\mathbf{O}_{gi}) - \varphi(\mathbf{O}_{gi})]) > t \right)$$

$$\leq \frac{\text{Var}\left( \sum_{g=1}^{G} \sum_{i=1}^{n_g} (\hat{\varphi}(\mathbf{O}_{gi}) - \varphi(\mathbf{O}_{gi})) \right)}{n^2 t^2}$$

$$= \frac{\sum_{g=1}^{G} \text{Var}\left( \sum_{i=1}^{n_g} (\hat{\varphi}(\mathbf{O}_{gi}) - \varphi(\mathbf{O}_{gi})) \right)}{n^2 t^2},$$

where the last equation follows from the independence of observations that come from different clusters. Set

$$t = \frac{M \sqrt{\sum_{g=1}^{G} \text{Var}\left( \sum_{i=1}^{n_g} (\hat{\varphi}(\mathbf{O}_{gi}) - \varphi(\mathbf{O}_{gi})) \right)}}{n},$$

we have

$$\mathbb{P}\left( \frac{1}{n} \sum_{g=1}^{G} \sum_{i=1}^{n_g} (\hat{\varphi}(\mathbf{O}_{gi}) - \varphi(\mathbf{O}_{gi}) - \mathbb{P}[\hat{\varphi}(\mathbf{O}_{gi}) - \varphi(\mathbf{O}_{gi})]) > t \right)$$

$$\leq \frac{1}{M^2}.$$

Thus the empirical process term can be bounded as

$$\frac{1}{n} \sum_{g=1}^{G} \sum_{i=1}^{n_g} (\hat{\varphi}(\mathbf{O}_{gi}) - \varphi(\mathbf{O}_{gi}) - \mathbb{P}[\hat{\varphi}(\mathbf{O}_{gi}) - \varphi(\mathbf{O}_{gi})])$$

$$= O_{\mathbb{P}}\left( \frac{\sqrt{\sum_{g=1}^{G} \text{Var}\left( \sum_{i=1}^{n_g} (\hat{\varphi}(\mathbf{O}_{gi}) - \varphi(\mathbf{O}_{gi})) \right)}}{n} \right).$$

For the third bias term, by the property of conditional expectation we have

$$\mathbb{P}[\hat{\varphi}(\mathbf{O}_{gi}) - \varphi(\mathbf{O}_{gi})]$$

$$= \mathbb{E}\left[ \frac{R_{gi}(Y_{gi} - \hat{\mu}(\mathbf{X}_g, \mathbf{W}_{gi}))}{\hat{\pi}(\mathbf{X}_g, \mathbf{W}_{gi})} + \hat{\mu}(\mathbf{X}_g, \mathbf{W}_{gi}) - \mu(\mathbf{X}_g, \mathbf{W}_{gi}) \right]$$

$$= \mathbb{E}\left[ \frac{\pi(\mathbf{X}_g, \mathbf{W}_{gi})(\mu(\mathbf{X}_g, \mathbf{W}_{gi}) - \hat{\mu}(\mathbf{X}_g, \mathbf{W}_{gi}))}{\hat{\pi}(\mathbf{X}_g, \mathbf{W}_{gi})} + \hat{\mu}(\mathbf{X}_g, \mathbf{W}_{gi}) - \mu(\mathbf{X}_g, \mathbf{W}_{gi}) \right]$$

$$= \mathbb{E}\left[ \frac{(\pi(\mathbf{X}_g, \mathbf{W}_{gi}) - \hat{\pi}(\mathbf{X}_g, \mathbf{W}_{gi}))(\mu(\mathbf{X}_g, \mathbf{W}_{gi}) - \hat{\mu}(\mathbf{X}_g, \mathbf{W}_{gi}))}{\hat{\pi}(\mathbf{X}_g, \mathbf{W}_{gi})} \right].$$

Since $\hat{\pi} \geq \epsilon$ and by Cauchy Schwarz inequality, we have

$$|\mathbb{P}[\hat{\varphi}(\mathbf{O}_{gi}) - \varphi(\mathbf{O}_{gi})]| \leq \frac{1}{\epsilon} \|\hat{\pi} - \pi\| \|\hat{\mu} - \mu\|,$$

which implies

$$|\mathbb{P}[\hat{\varphi}(\mathbf{O}_{gi}) - \varphi(\mathbf{O}_{gi})]| = O_{\mathbb{P}}(\|\hat{\pi} - \pi\| \|\hat{\mu} - \mu\|).$$

This completes the proof of the asymptotic expansion. The asymptotic normality then follows from Slutsky's theorem.

# G   Proof of Corollary 1

The conditional variance given the training set $D$ can be expressed as

$$\sum_{g=1}^{G} \text{Var}\left(\sum_{i=1}^{n_g}(\hat{\varphi}(\mathbf{O}_{gi}) - \varphi(\mathbf{O}_{gi}))\right)$$

$$= \sum_{g=1}^{G} \sum_{i,j} \text{Cov}\left(\hat{\varphi}(\mathbf{O}_{gi}) - \varphi(\mathbf{O}_{gi}), \hat{\varphi}(\mathbf{O}_{gj}) - \varphi(\mathbf{O}_{gj})\right)$$

$$\leq \sum_{g=1}^{G} \sum_{i,j} \text{Var}\left(\hat{\varphi}(\mathbf{O}_{gi}) - \varphi(\mathbf{O}_{gi})\right)$$

$$= \sum_{g=1}^{G} n_g^2 \text{Var}\left(\hat{\varphi}(\mathbf{O}) - \varphi(\mathbf{O})\right)$$

$$\leq \sum_{g=1}^{G} n_g^2 \|\hat{\varphi}(\mathbf{O}) - \varphi(\mathbf{O})\|^2,$$

where we have used Cauchy Schwarz inequality to bound the covariance with variance and the fact that the observations $\{\mathbf{O}_{gi}, 1 \leq i \leq n_g, 1 \leq g \leq G\}$ are identically distributed.

# H   Proof of Theorem 2

We have the following error decomposition

$$\hat{\psi}_{DR} - \psi_n = \frac{1}{n}\sum_{g=1}^{G}\sum_{t=1}^{n_g}(\varphi(\mathbf{Z}_{gt}) - \mathbb{E}[\varphi(\mathbf{Z}_{gt})])$$

$$+ \frac{1}{n}\sum_{g=1}^{G}\sum_{t=1}^{n_g}(\hat{\varphi}(\mathbf{Z}_{gt}) - \varphi(\mathbf{Z}_{gt}) - \mathbb{P}[\hat{\varphi}(\mathbf{Z}_{gt}) - \varphi(\mathbf{Z}_{gt})])$$

$$+ \frac{1}{n}\sum_{g=1}^{G}\sum_{t=1}^{n_g}\mathbb{P}[\hat{\varphi}(\mathbf{Z}_{gt}) - \varphi(\mathbf{Z}_{gt})],$$

where note that $\{\mathbf{Z}_{gt}, 1 \leq t \leq n_g, 1 \leq g \leq G,\}$ may not share the same distribution in general. The empirical process can be bounded using the same technique as in the proof of Theorem 1:

$$\frac{1}{n}\sum_{g=1}^{G}\sum_{t=1}^{n_g}(\hat{\varphi}(\mathbf{Z}_{gt}) - \varphi(\mathbf{Z}_{gt}) - \mathbb{P}[\hat{\varphi}(\mathbf{Z}_{gt}) - \varphi(\mathbf{Z}_{gt})])$$

$$= O_{\mathbb{P}}\left(\frac{\sqrt{\sum_{g=1}^{G}\text{Var}\left(\sum_{t=1}^{n_g}\hat{\varphi}(\mathbf{Z}_{gt}) - \varphi(\mathbf{Z}_{gt}) \mid D\right)}}{n}\right)$$

since independence still holds across clusters. For the conditional bias term, we have (conditioning on the training set D)

$$\mathbb{P}[\hat{\varphi}(\mathbf{Z}_{gt}) - \varphi(\mathbf{Z}_{gt})]$$

$$= \mathbb{E}\left[\frac{R_{gt}(Y_{gt} - \hat{\mu}(\mathbf{X}_g, \mathbf{S}_{gt}))}{\hat{\pi}(\mathbf{X}_g, \mathbf{S}_{gt})} + \hat{\mu}(\mathbf{X}_g, \mathbf{S}_{gt}) - \mu(\mathbf{X}_g, \mathbf{S}_{gt})\right]$$

$$= \mathbb{E}\left[\frac{\pi_{gt}(\mathbf{X}_g, \mathbf{H}_{gt})(\mu_{gt}(\mathbf{X}_g, \mathbf{H}_{gt}) - \hat{\mu}(\mathbf{X}_g, \mathbf{S}_{gt}))}{\hat{\pi}(\mathbf{X}_g, \mathbf{S}_{gt})} + \hat{\mu}(\mathbf{X}_g, \mathbf{S}_{gt}) - \mu(\mathbf{X}_g, \mathbf{S}_{gt})\right]$$

$$= \mathbb{E}\left[\frac{\pi(\mathbf{X}_g, \mathbf{S}_{gt})(\mu(\mathbf{X}_g, \mathbf{S}_{gt}) - \hat{\mu}(\mathbf{X}_g, \mathbf{S}_{gt}))}{\hat{\pi}(\mathbf{X}_g, \mathbf{S}_{gt})} + \hat{\mu}(\mathbf{X}_g, \mathbf{S}_{gt}) - \mu(\mathbf{X}_g, \mathbf{S}_{gt})\right]$$

$$= \mathbb{E}\left[\frac{(\pi(\mathbf{X}_g, \mathbf{S}_{gt}) - \hat{\pi}(\mathbf{X}_g, \mathbf{S}_{gt}))(\mu(\mathbf{X}_g, \mathbf{S}_{gt}) - \hat{\mu}(\mathbf{X}_g, \mathbf{S}_{gt}))}{\hat{\pi}(\mathbf{X}_g, \mathbf{S}_{gt})}\right],$$

where the second equation follows from conditioning on $(\mathbf{X}_g, \mathbf{H}_{gt})$ and the third equation follows from Assumption 3. Hence we have

$$\left| \frac{1}{n} \sum_{g=1}^{G} \sum_{t=1}^{n_g} \mathbb{P}[\hat{\varphi}(\mathbf{Z}_{gt}) - \varphi(\mathbf{Z}_{gt})] \right| \tag{7}$$
$$\leq \frac{1}{\epsilon n} \sum_{g=1}^{G} \sum_{t=1}^{n_g} \|\hat{\pi}(\mathbf{X}_g, \mathbf{S}_{gt}) - \pi(\mathbf{X}_g, \mathbf{S}_{gt})\| \|\hat{\mu}(\mathbf{X}_g, \mathbf{S}_{gt}) - \mu(\mathbf{X}_g, \mathbf{S}_{gt})\|.$$

Thus the conditional bias term can be bounded as

$$\frac{1}{n} \sum_{g=1}^{G} \sum_{t=1}^{n_g} \mathbb{P}[\hat{\varphi}(\mathbf{Z}_{gt}) - \varphi(\mathbf{Z}_{gt})])$$
$$= O_{\mathbb{P}}\left( \frac{1}{n} \sum_{g=1}^{G} \sum_{t=1}^{n_g} \|\hat{\pi}(\mathbf{X}_g, \mathbf{S}_{gt}) - \pi(\mathbf{X}_g, \mathbf{S}_{gt})\| \|\hat{\mu}(\mathbf{X}_g, \mathbf{S}_{gt}) - \mu(\mathbf{X}_g, \mathbf{S}_{gt})\| \right).$$

The asymptotic expansion is then proved. The asymptotic normality then follows from Hansen and Lee (2019)[Theorem 2] and Slutsky's theorem.

# I   Proof of Corollary 2

First, to bound the empirical process term, the conditional variance given the training set $D$ can be expressed as

$$\sum_{g=1}^{G} \mathrm{Var}\left( \sum_{i=1}^{n_g} (\hat{\varphi}(\mathbf{Z}_{gi}) - \varphi(\mathbf{Z}_{gi})) \mid D \right)$$
$$= \sum_{g=1}^{G} \sum_{i,j} \mathrm{Cov}\left( \hat{\varphi}(\mathbf{Z}_{gi}) - \varphi(\mathbf{Z}_{gi}), \hat{\varphi}(\mathbf{Z}_{gj}) - \varphi(\mathbf{Z}_{gj}) \mid D \right)$$
$$\leq \sum_{g=1}^{G} \sum_{i,j} \sqrt{\mathrm{Var}\left( \hat{\varphi}(\mathbf{Z}_{gi}) - \varphi(\mathbf{Z}_{gi}) \mid D \right) \mathrm{Var}\left( \hat{\varphi}(\mathbf{Z}_{gj}) - \varphi(\mathbf{Z}_{gj}) \mid D \right)}$$
$$\leq \sum_{g=1}^{G} \sum_{i,j} \sqrt{\mathbb{E}_{\mathbf{Z}_{gi}}[\hat{\varphi}(\mathbf{Z}_{gi}) - \varphi(\mathbf{Z}_{gi})^2] \mathbb{E}_{\mathbf{Z}_{gj}}[\hat{\varphi}(\mathbf{Z}_{gj}) - \varphi(\mathbf{Z}_{gj})^2]}$$

where we have used Cauchy Schwarz inequality to bound the covariance with variance and $\mathbb{E}_{\mathbf{Z}_{gi}}$ means the expectation is taken over $\mathbf{Z}_{gi}$. Now we have

$$\mathbb{E}_D\left[ \sum_{g=1}^{G} \mathrm{Var}\left( \sum_{i=1}^{n_g} (\hat{\varphi}(\mathbf{Z}_{gi}) - \varphi(\mathbf{Z}_{gi})) \mid D \right) \right]$$
$$\leq \sum_{g=1}^{G} \sum_{i,j} \mathbb{E}_D\left[ \sqrt{\mathbb{E}_{\mathbf{Z}_{gi}}[(\hat{\varphi}(\mathbf{Z}_{gi}) - \varphi(\mathbf{Z}_{gi}))^2] \mathbb{E}_{\mathbf{Z}_{gj}}[(\hat{\varphi}(\mathbf{Z}_{gj}) - \varphi(\mathbf{Z}_{gj}))^2]} \right]$$
$$\leq \sum_{g=1}^{G} \sum_{i,j} \left[ \sqrt{\mathbb{E}_D\left[ \mathbb{E}_{\mathbf{Z}_{gi}}(\hat{\varphi}(\mathbf{Z}_{gi}) - \varphi(\mathbf{Z}_{gi}))^2 \right] \mathbb{E}_D\left[ \mathbb{E}_{\mathbf{Z}_{gj}}((\hat{\varphi}(\mathbf{Z}_{gj}) - \varphi(\mathbf{Z}_{gj}))^2) \right]} \right]$$
$$= \sum_{g=1}^{G} \sum_{i,j} \left[ \sqrt{\mathbb{E}_{\mathbf{Z}_{gi}}\left[ \mathbb{E}_D(\hat{\varphi}(\mathbf{Z}_{gi}) - \varphi(\mathbf{Z}_{gi}))^2 \right] \mathbb{E}_{\mathbf{Z}_{gj}}\left[ \mathbb{E}_D(\hat{\varphi}(\mathbf{Z}_{gj}) - \varphi(\mathbf{Z}_{gj}))^2 \right]} \right]$$
$$\leq \sum_{g=1}^{G} \sum_{i,j} \sup_{\mathbf{z}} \mathbb{E}_D\left[ (\hat{\varphi}(\mathbf{z}) - \varphi(\mathbf{z}))^2 \right]$$
$$= \sum_{g=1}^{G} n_g^2 \mathbb{E}_D\left[ (\hat{\varphi}(\mathbf{z}) - \varphi(\mathbf{z}))^2 \right].$$

Thus we have

$$\frac{1}{n}\sqrt{\sum_{g=1}^{G}\text{Var}\left(\sum_{i=1}^{n_g}(\hat{\varphi}(\mathbf{Z}_{gi})-\varphi(\mathbf{Z}_{gi}))\mid D\right)}=O_{\mathbb{P}}\left(\frac{1}{n}\sqrt{\sum_{g=1}^{G}n_g^2\sup_{\mathbf{z}}\mathbb{E}_D\left[\hat{\varphi}(\mathbf{z})-\varphi(\mathbf{z})^2\right]}\right).$$

For the bias term, from equation (7) the proof of Theorem 2 we have

$$\mathbb{E}_D\left[\left|\left|\frac{1}{n}\sum_{g=1}^{G}\sum_{t=1}^{n_g}\mathbb{P}[\hat{\varphi}(\mathbf{Z}_{gt})-\varphi(\mathbf{Z}_{gt})]\right|\right|\right]$$

$$\leq\frac{1}{\epsilon n}\sum_{g=1}^{G}\sum_{t=1}^{n_g}\mathbb{E}_D\left[\|\hat{\pi}(\mathbf{X}_g,\mathbf{S}_{gt})-\pi(\mathbf{X}_g,\mathbf{S}_{gt})\|\|\hat{\mu}(\mathbf{X}_g,\mathbf{S}_{gt})-\mu(\mathbf{X}_g,\mathbf{S}_{gt})\|\right]$$

$$\leq\frac{1}{\epsilon n}\sum_{g=1}^{G}\sum_{t=1}^{n_g}\sqrt{\mathbb{E}_D\left[\|\hat{\pi}(\mathbf{X}_g,\mathbf{S}_{gt})-\pi(\mathbf{X}_g,\mathbf{S}_{gt})\|^2\right]\mathbb{E}_D\left[\|\hat{\mu}(\mathbf{X}_g,\mathbf{S}_{gt})-\mu(\mathbf{X}_g,\mathbf{S}_{gt})\|^2\right]}$$

$$=\frac{1}{\epsilon n}\sum_{g=1}^{G}\sum_{t=1}^{n_g}\sqrt{\mathbb{E}_{\mathbf{X}_g,\mathbf{S}_{gt}}\left[\mathbb{E}_D(\hat{\pi}(\mathbf{X}_g,\mathbf{S}_{gt})-\pi(\mathbf{X}_g,\mathbf{S}_{gt}))^2\right]\mathbb{E}_{\mathbf{X}_g,\mathbf{S}_{gt}}\left[\mathbb{E}_D(\hat{\mu}(\mathbf{X}_g,\mathbf{S}_{gt})-\mu(\mathbf{X}_g,\mathbf{S}_{gt}))^2\right]}$$

$$\leq\frac{1}{\epsilon n}\sum_{g=1}^{G}\sum_{t=1}^{n_g}\sqrt{\sup_{\boldsymbol{x},\boldsymbol{s}}\mathbb{E}_D(\hat{\pi}(\boldsymbol{x},\boldsymbol{s})-\pi(\boldsymbol{x},\boldsymbol{s}))^2\sup_{\boldsymbol{x},\boldsymbol{s}}\mathbb{E}_D(\hat{\mu}(\boldsymbol{x},\boldsymbol{s})-\mu(\boldsymbol{x},\boldsymbol{s}))^2}$$

$$=\frac{1}{\epsilon}\sqrt{\sup_{\boldsymbol{x},\boldsymbol{s}}\mathbb{E}_D(\hat{\pi}(\boldsymbol{x},\boldsymbol{s})-\pi(\boldsymbol{x},\boldsymbol{s}))^2\sup_{\boldsymbol{x},\boldsymbol{s}}\mathbb{E}_D(\hat{\mu}(\boldsymbol{x},\boldsymbol{s})-\mu(\boldsymbol{x},\boldsymbol{s}))^2}.$$

Thus we conclude

$$\frac{1}{n}\sum_{g=1}^{G}\sum_{t=1}^{n_g}\|\hat{\pi}(\mathbf{X}_g,\mathbf{S}_{gt})-\pi(\mathbf{X}_g,\mathbf{S}_{gt})\|\|\hat{\mu}(\mathbf{X}_g,\mathbf{S}_{gt})-\mu(\mathbf{X}_g,\mathbf{S}_{gt})\|$$

$$=O_{\mathbb{P}}\left(\sqrt{\sup_{\boldsymbol{x},\boldsymbol{s}}\mathbb{E}_D(\hat{\pi}(\boldsymbol{x},\boldsymbol{s})-\pi(\boldsymbol{x},\boldsymbol{s}))^2\sup_{\boldsymbol{x},\boldsymbol{s}}\mathbb{E}_D(\hat{\mu}(\boldsymbol{x},\boldsymbol{s})-\mu(\boldsymbol{x},\boldsymbol{s}))^2}\right),$$

