# OpenReview forum: "Handling Missing Responses under Cluster Dependence with Applications to Language Model Evaluation"
_NeurIPS.cc/2025/Conference — NeurIPS 2025 poster_

### Official Review · Reviewer_XqtA · 2025-07-02

**Clarity:** 3
**Significance:** 3
**Originality:** 3
**Rating:** 4
**Confidence:** 1

**Summary:**

This paper addresses the challenge of evaluating language models using human feedback, where missing annotations and cluster dependence often occur. It analyzes the doubly robust estimator—widely used in missing data analysis and causal inference—and extends its theoretical properties under clustered data. The authors establish new asymptotic results, showing how accounting for within-cluster correlation leads to faster convergence rates. Through simulations and real-world datasets, they demonstrate that their approach enables more reliable statistical inference when handling missing responses and clustered structures.

**Questions:**

1. The paper does not sufficiently discuss the limitations of the proposed method or provide insights into future research directions and its potential impact on the broader research community.

**Ethical Concerns:**

["NO or VERY MINOR ethics concerns only"]

**Final Justification:**

Thank you for the detailed and thoughtful rebuttal. I appreciate the authors’ efforts in addressing the concerns and clarifying key points. Although I have not thoroughly verified the mathematical derivations in the paper, the problem formulation and modeling appear clear and reasonable.

***However, because this topic is outside my core area of expertise, I will retain my original assessment and leave the final judgment to the area chair.***

**Limitations:**

Although the theoretical analysis is solid, the paper lacks validation on large-scale real-world datasets.

**Paper Formatting Concerns:**

There are no formatting concerns.

**Quality:**

3

**Strengths And Weaknesses:**

### Strengths

1. The paper provides a clear formulation and notation, which helps readers quickly understand the problem setup and modeling assumptions.

2. It systematically considers missing data under both simultaneous sampling and sequential sampling scenarios, with precise definitions and mathematical modeling for each case.

3. The proposed method is thoroughly validated through both simulation studies and real-world data analysis, demonstrating its effectiveness and robustness in practical applications.

### Weakness
1. Although the theoretical analysis is solid, the paper lacks validation on large-scale real-world datasets.

---

> ### Author Rebuttal · Authors · 2025-07-29
>
> We thank the reviewer for the thoughtful and constructive comments, insightful summary of our key contributions, and positive feedback on the clarity, organization, and significance of our work. These suggestions have been valuable in improving the paper. Our responses to the weaknesses and questions are as follows:
>
> **Weaknesses:**
>
> We agree that empirical validation is important. Our initial submission indeed
> included a real-data example in Section 6 to illustrate our methods (although not a large one, due to access limitations). The results highlight the importance of accounting for cluster dependence in multiturn LLM evaluation: confidence intervals based on the cluster-robust variance estimator successfully cover the true average for all outcomes of interest, whereas those that ignore the dependence structure fail when the outcome $Y$ is humor. This underscores the necessity of incorporating the dependence structure when analyzing multiturn LLM evaluation data.
> As part of the rebuttal, we also add additional empirical experiments to demonstrate the advantages of our approach, including its double robustness—specifically, the consistency of the DR estimator when either nuisance function is correctly specified.
>
> **Questions:**
>
> In the submission, we have highlighted several limitations of our work. For instance, as noted in the discussion following Theorem 1, practitioners often rely on i.i.d-based methods to estimate nuisance functions in clustered settings; however, the theoretical justification for this practice remains an open question. Additionally, at the end of Section 3, we mentioned that while cluster bootstrap methods can be used for variance estimation, they tend to be computationally intensive. Finally, Assumption 1 requires the existence of a summary variable, $S_{gt}$, and we emphasize that its specification is context-dependent and relies on domain knowledge. Finally, our approach relies on the MAR assumption—that is, all variables influencing the missingness mechanism are observed and included in the analysis. When this assumption may not hold, sensitivity analysis is necessary to assess the robustness of the results.
>
> In the original version, we discussed two directions for future work: addressing instability when the propensity score is close to zero—potentially by incorporating balancing weights—and learning human annotation rules from labeled datasets to enable evaluation of LLMs without human labels. In this revision, we have added a third direction: developing nuisance estimation algorithms for dependent data and establishing their theoretical guarantees.
>
> Regarding the impact on the research community, we believe our contributions are twofold. From the perspective of the causal inference literature, our theoretical results provide justification for using doubly robust estimators in the analysis of clustered data, especially when the cluster sizes are unbounded. In the context of LLM evaluation, the proposed method is novel in its ability to handle tasks with missing human annotations. In particular, the idea of incorporating summaries of historical information into the estimation procedure has important applications in multiturn LLM evaluation. We have added the following discussion to the introduction to highlight these contributions.
>
> "Our results provide theoretical justification for using doubly robust estimators in the analysis of clustered data, especially when the cluster sizes are unbounded. The proposed methods, such as incorporating summaries of historical information into the estimation procedure, have important applications in multiturn LLM evaluation with missing human annotations, as demonstrated in our real-data example."

---

> > ### Comment · Reviewer_XqtA · 2025-08-05
> >
> > Thank you for the detailed and thoughtful rebuttal. I appreciate the authors' efforts in addressing the concerns and clarifying key points. ***However, given my limited expertise in this specific area, I will maintain my original review and defer the final decision to the area chair.***

---

### Official Review · Reviewer_564S · 2025-07-02

**Clarity:** 3
**Significance:** 3
**Originality:** 2
**Rating:** 4
**Confidence:** 3

**Summary:**

This paper addresses the problem of estimating average outcomes from human annotations in settings where data points are clustered (e.g. multiple responses from the same user) and some outcome values are missing. The authors focus on the doubly robust estimator and investigate its properties under cluster-dependent data. The authors derive theoretical results showing that the estimator remains asymptotically unbiased and normal under cluster sampling, even allowing cluster sizes to grow with the sample. The paper includes simulation studies demonstrating the importance of accounting for cluster structure. The paper also includes a real-world case study on the OpenAssistant conversation dataset, where the method is applied to estimate average conversation quality metrics with simulated missing ratings. The adjusted estimates with cluster-robust inference are shown to correct the bias of naive estimates and provide valid confidence intervals. Overall, the main contributions of this paper are a theoretical extension of doubly robust estimation to clustered and sequential settings, and empirical evidence underlining the need for cluster-aware inference in human feedback evaluation.

**Questions:**

1. Could the authors provide more guidance or discussion on choosing the summary variable $S_{gt}$ for sequential data?
2. The paper uses a cluster robust variance estimator and mentions cluster-level bootstrap as a more robust alternative. Have the authors considered using the wild bootstrap or other resampling techniques specifically designed for cluster dependence?
3. While the doubly robust estimator has great properties, some readers might wonder how it compares to simpler approaches, such as complete-case analysis (ignore missing), mean imputation, or even model-based imputation.

**Ethical Concerns:**

["NO or VERY MINOR ethics concerns only"]

**Limitations:**

Yes

**Quality:**

3

**Strengths And Weaknesses:**

Strengths:
1. The authors extend the theory of doubly robust estimation to clustered data with possibly unbounded cluster sizes. They show the estimator is asymptotically normal with a variance inflation that depends on within-cluster correlation.
2. The use of the doubly robust estimator is appropriate as it combines outcome modeling and propensity weighting, which offers protection against misspecification of either component. The paper proves the estimator’s double robustness and rate-adaptive properties in the clustered setting.
3. The paper tackles temporal dependence within clusters which is very relevant for multi-turn interactions in conversational language models, through using summary statistics from conversation history.

Weaknesses:
1. While the theoretical extension to unbounded cluster sizes and sequential dependence is important, some aspects may be incremental. The doubly robust estimator itself is not new, and previous works (e.g. Yang 2018; Park & Kang 2021) have studied missing data or treatment effects in clustered settings. The contribution is more on the theoretical side than a new algorithm.
2. The analysis relies on the missing at random assumption. If there are unobserved reasons why some responses are missing, the estimator could still be biased. The paper does not explore sensitivity to potential missing not at random scenarios, which could be relevant in real human feedback, e.g. users might skip rating bad responses.
3. In the OpenAssistant dataset experiment, the missingness was simulated by the authors. While this is fine to illustrate the method, it would be even more convincing to apply the approach on a dataset with naturally occurring missing responses or non-response bias.
4. The evaluation focuses on point estimates and CIs of the mean. It does not compare to alternative missing data techniques, such as multiple imputation or simple mean imputation.

---

> ### Author Rebuttal · Authors · 2025-07-29
>
> We thank the reviewer for the thoughtful and constructive comments, insightful summary of our key contributions, and positive feedback on the clarity, organization, and significance of our work. These suggestions have been valuable in improving the paper. Our responses to the weaknesses and questions are as follows:
>
>
> **Weaknesses:**
> 1. We acknowledge that the doubly robust estimator itself is not new, and Section 3 focuses on examining its theoretical properties in clustered settings. However, a key contribution of this work also lies in Section 4, where we formalize the idea of incorporating summaries of historical information into the estimation procedure in the sequential setting—a novel approach with important applications in multiturn LLM evaluation.
> 2. We certainly appreciate the point that the missing at random (MAR) assumption can sometimes be strong. First, we want to emphasize that MAR is actually quite flexible in the scenarios we consider. For instance, the example provided by the reviewer---such as users being more likely to skip rating poor responses---can still fall within the MAR framework. In such cases, the missingness depends on the content of the system's response, which is captured by the variables $W_{gi}$ in our work. Since both our outcome model and propensity score model adjust for $W_{gi}$, our methods are capable of handling missing feedback that arises from poor system responses, as this information is explicitly incorporated into the estimation and adjustment procedures. So the MAR framework includes a wide range of practical scenarios. However, if unobserved variables such as some user information
>  also affects the missingness, the MAR assumption will be violated. Second, we agree with the reviewer that a sensitivity analysis approach would be promising here. While a complete exploration of this idea is outside the scope of the current paper, we can readily combine our novel results on doubly estimation with several popular sensitivity analysis frameworks. We now discuss this briefly after Assumption 1:
>
> "When unobserved confounders may influence the missing mechanism, the MAR assumption may no longer hold. To assess the sensitivity of our results to such violations, we can follow the framework of Cinelli and Hazlett (2020), which extends classical omitted variable bias analysis. This approach quantifies the strength that an unobserved confounder would need to exhibit—measured by its partial $R^2$ with both the treatment or missingness indicator and the outcome—to reduce the estimated effect to zero or render it statistically insignificant. The robustness value (RV) summarizes this threshold and can be benchmarked against observed covariates for interpretation."
>
> 3. We agree with the reviewer that applying our approach to a dataset with naturally occurring missing responses would provide a more compelling demonstration. And while we have been actively exploring a range of different applications, publicly available data are unfortunately quite rare.
>     Given those constraints, we believe that simulating the missingness also offers important advantages. In particular, it allows us to treat the full-sample average as the "ground truth," enabling a clear evaluation of our methods---such as assessing whether the resulting confidence intervals achieve the desired coverage.
>
> 4. See our response to Q4 for details.
>
> **Questions:**
> 1. The choice of summary variable $S_{gt}$ depends on specific applications and often requires domain knowledge. Common choices include average or cumulative measures of historical information. We have added more discussion after Assumption 3 as follows:
>
> "The variable $S_{gt}$ can be viewed as a sufficient summary of the historical information up to time $t$. In practice, the choice of $S_{gt}$ often requires domain knowledge. Common choices include average or cumulative measures of past information. For example, in mobile health studies, a user wears a fitness tracker that collects data daily. The device may fail to record data at time $t$ due to battery depletion, which depends on historical usage; in this case, $S_{gt}$ could represent the cumulative device usage over the past few days. In educational testing or tutoring systems, whether a student attempts a question at time $t$ may depend on cumulative difficulty or frustration from earlier interactions. Here, $S_{gt}$ might be defined as the number of incorrect attempts or a difficulty-adjusted score accumulated up to time $t$. In the LLM evaluation setting, whether users provide feedback may depend on their prior interactions with the system. Accordingly, $S_{gt}$ can be constructed as the embedding of the concatenated conversation history $\overline{W}_{gt}$ up to time $t$, or from the most recent $d$ conversations $W\_{g,t-d+1}, \dots, W\_{gt}$. These embeddings remain of fixed length regardless of the length of the conversation history."
>
> 2. Thank you for pointing out additional resampling methods for variance estimation. In addition to the bootstrap approaches for clustered data discussed after Equation (4), we have now included further discussion on the use of cluster wild bootstrap, which is particularly useful in settings with heteroskedasticity or within-cluster dependence:
>
> "An alternative is the cluster wild bootstrap, which is well-suited for settings with heteroskedasticity, few
> clusters, or varying cluster sizes (MacKinnon and Webb, 2017)."
>
> 3. This is a good point, and we agree that expanding our comparison with simpler approaches is important.
> In both our simulations and real-data analysis, we now include complete-case analysis---which ignores missing data---as a baseline. As expected, it yields biased estimates under the MAR setting because it fails to account for the missingness mechanism. Similarly, simple mean imputation is also biased. To compare the doubly robust estimator with model-based imputation, we note that the doubly robust approach can be viewed as a form of imputation using the "pseudo-outcome" or influence function $\varphi$ defined in Theorem 1. In contrast, model-based imputation typically relies on regression and corresponds to the outcome regression estimator, which models only $\mu$ and is consistent only if the outcome model is correctly specified. The doubly robust estimator combines both the propensity score and the outcome model, and remains consistent if either is correctly specified. In this sense, it provides a more robust imputation strategy. Moreover, when both models are correctly specified, it achieves semiparametric efficiency. Overall, the doubly robust estimator offers clear advantages over simpler alternatives. We have added the following sentence after the plug-in estimator on line 130 to connect it with regression-based imputation:
>
> "This estimator corresponds to regression-based imputation and is consistent (under mild conditions)
> when $\hat{\mu}$ is consistent. However, the plug-in-style estimator usually suffers from first-order bias and is
> not robust to model misspecification (Bang and Robins, 2005; Funk et al., 2011).."

---

### Official Review · Reviewer_XGKK · 2025-07-05

**Clarity:** 2
**Significance:** 3
**Originality:** 2
**Rating:** 4
**Confidence:** 2

**Summary:**

This paper addresses the problem of estimating population means under missing outcome data with cluster-dependent observations, common in applications like LLM evaluation or education studies. Key contributions include:
1.Theoretical guarantees for the DR estimator under cluster dependence, allowing growing cluster sizes and establishing convergence rates;
2. A novel use of summary variables to simplify nuisance function modeling in sequential settings;
3.Simulation and real-data studies (on OpenAssistant) demonstrating improved coverage and robustness over i.i.d.-based methods.

**Questions:**

Nuisance Function Estimation
Please clarify practical methods for estimating propensity scores and outcome models in clustered/sequential data, especially with varying cluster sizes. Concrete examples or recommended estimation procedures would strengthen the practical impact.
Robustness to Misspecification
Discuss how the estimator performs if both nuisance models are misspecified in finite samples under cluster dependence. The doubly robust estimator is known for consistency when either nuisance model is correct. Could the authors discuss the estimator’s robustness in finite samples under simultaneous misspecification of both nuisance models, especially in clustered or dependent data contexts? Empirical or theoretical insights would help readers understand limitations.
Effect of Large Clusters
Elaborate on the impact of very large cluster sizes on convergence rates and estimator performance, with intuition or examples.

**Ethical Concerns:**

["NO or VERY MINOR ethics concerns only"]

**Final Justification:**

thanks for the response from authors and i would like to keep my original rating.

**Limitations:**

yes

**Quality:**

3

**Strengths And Weaknesses:**

Strengths
Quality: Rigorous theoretical development extending doubly robust estimation to clustered and sequential dependent data, with asymptotic normality and convergence rates. Supported by simulation and real data analysis.
Clarity: Well-structured presentation with clear assumptions and theorems. Practical examples (e.g., LLM evaluation) help illustrate the methodology.
Significance: Addresses important problems in missing data under cluster dependence, allowing diverging cluster sizes, which is relevant for modern complex data structures.
Originality: Novel relaxation of bounded cluster size assumptions and formal incorporation of historical summary variables in sequential missingness models.

Weaknesses
Quality: Strong assumptions on model correctness and independent training samples; robustness to model misspecification could be further explored. Variance estimation may be challenging in practice.
Clarity: Dense notation and technical conditions may limit accessibility; limited guidance on choosing or constructing summary variables for sequential data.
Significance: Focuses mainly on mean estimation, leaving other causal parameters unexplored; computational considerations for large datasets are not discussed in depth.
Originality: Use of summary variables aligns with existing dimension reduction ideas; core doubly robust methods in clustered missing data are established, so contribution is primarily theoretical extension.

---

> ### Author Rebuttal · Authors · 2025-07-29
>
> We thank the reviewer for the thoughtful and constructive comments, insightful summary of
> our key contributions, and positive feedback on the clarity, organization, and significance of our work. These
> suggestions have been valuable in improving the paper. Our responses to the weaknesses and questions are as follows:
>
>
> **Weaknesses:**
>
> 1. While the assumption of an independent training sample might be strong in some applications,
> we argue this can often be easy to satisfy in practice via sample splitting and cross-fitting (Chernozhukov
> et al., 2018) to retain full-sample efficiency. For a discussion on model specification, see our response to Q2.
>
> 2. We appreciate this point and have revised the submission to improve both presentation and practical guidance in a number of points. The choice of summary variable $S_{gt}$ depends on specific applications and often requires domain knowledge. Common choices include average or cumulative measures of historical information. We have added more discussion after Assumption 3 as follows:
>
> "The variable $S_{gt}$ can be viewed as a sufficient summary of the historical information up to time $t$. In practice, the choice of $S_{gt}$ often requires domain knowledge. Common choices include average or cumulative measures of past information. For example, in mobile health studies, a user wears a fitness tracker that collects data daily. The device may fail to record data at time $t$ due to battery depletion, which depends on historical usage; in this case, $S_{gt}$ could represent the cumulative device usage over the past few days. In educational testing or tutoring systems, whether a student attempts a question at time $t$ may depend on cumulative difficulty or frustration from earlier interactions. Here, $S_{gt}$ might be defined as the number of incorrect attempts or a difficulty-adjusted score accumulated up to time $t$. In the LLM evaluation setting, whether users provide feedback may depend on their prior interactions with the system. Accordingly, $S_{gt}$ can be constructed as the embedding of the concatenated conversation history $\overline{W}_{gt}$ up to time $t$, or from the most recent $d$ conversations $W\_{g,t-d+1}, \dots, W\_{gt}$. These embeddings remain of fixed length regardless of the length of the conversation history."
>
> **Questions:**
>
> 1. The nuisance estimation problem is essentially a regression problem. In the initial
> submission, we discussed several regression methods suitable for clustered data in the paragraph
> beginning at line 167, including multilevel GLM, kernel regression, random forests and neural networks.
> The recommended approach depends on the specific objective. For instance, if interpretability is
> important, multilevel generalized linear models (GLMs) are preferred. If the goal is to avoid strong
> parametric assumptions, nonparametric methods such as kernel regression (Shimizu, 2024) or random
> forests (Young and Buhlmann, 2025) are recommended. Researchers can choose approaches to estimate
> the nuisance functions based on domain knowledge and their objectives in practice.
>
> 2. Theorem 1 implies the same consistency guarantees hold for the DR estimator in the clustered setting, i.e., it is consistent when either nuisance model is correctly specified. We include additional simulation studies in the appendix to compare the regression estimator, IPW estimator and DR estimator in the clustered setting under potential model misspecification to illustrate the advantage of the DR estimator as follows:
>
> "In this section, we present additional simulation results for the plug-in (regression) estimator
> $$
> \hat{\theta}_{\mathrm{OR}} = \frac{1}{n} \sum\_{g=1}^G \sum\_{i=1}^{n\_g} \hat{\mu}(X\_g, W\_{gi}),
> $$
>
> the IPW estimator
> $$
> \hat{\theta}_{\text{IPW}} = \frac{1}{n} \sum\_{g=1}^G \sum\_{i=1}^{n\_g} \frac{R\_{gi}Y\_{gi}}{\hat{\pi}(X\_g, W\_{gi})},
> $$
> and the doubly robust estimator in equation (2). Our focus is on evaluating their performance under model misspecification in the presence of clustered data.
>
> The data-generating process follows the homogeneous sampling setup described in Appendix C, with the regression function and propensity score given by
> $$
> \mu(X\_g, W\_{gi}) = -X\_g + W\_{gi}^2, \quad \pi(X\_g, W\_{gi}) = \text{logistic}(X\_g + 0.5W\_{gi}^2).
> $$
> The true average outcome is $ \theta = 5 $. To assess estimator performance under misspecification, we consider scenarios in which $ \mu $ and/or $ \pi $ are misspecified by modeling the quadratic term in $ W\_{gi} $ as linear.
> We generate samples of size $ n= 1000, 10000 $ with varying cluster sizes, apply each estimator under different model specifications, and compute the mean squared error (MSE). The results are summarized in the following table.
>
> **Table: Mean squared error (MSE) of regression, IPW, and DR estimators under potential nuisance misspecification with sample size $n = 10000$, $n_g = 100$.**
>
> | Scenario                        | Regression Estimator | IPW Estimator | DR Estimator |
> |--------------------------------|----------------------|---------------|--------------|
> | $\mu$ correct, $\pi$ correct   | 0.0562               | 0.0561        | 0.0563       |
> | $\mu$ correct, $\pi$ wrong     | 0.0563               | 2.7023        | 0.0563       |
> | $\mu$ wrong, $\pi$ correct     | 2.0356               | 0.0563        | 0.0572       |
> | $\mu$ wrong, $\pi$ wrong       | 2.0603               | 2.6896        | 2.5765       |
>
> The conclusions in this clustered setting are similar to those in the classical i.i.d. setting. When the outcome model $ \mu $ is misspecified, the plug-in (regression) estimator $ \hat{\theta}\_{\text{OR}} $ becomes inconsistent. Similarly, the consistency of the IPW estimator $ \hat{\theta}\_{\text{IPW}} $ relies on correct specification of the propensity score $ \pi $. In contrast, the doubly robust estimator $ \hat{\theta}\_{\text{DR}} $, which models both the outcome and the missingness mechanism, remains consistent as long as either $ \mu $ or $ \pi $ is correctly specified. This aligns with the theoretical guarantees established in Theorem 1."
>
> We also include two additional tables with different sample and cluster sizes. The results are similar to the table presented above.
>
> 3. We have added an additional example in Appendix B to illustrate how heterogeneous cluster sizes can affect convergence rates:
>
> "Consider a setting with two types of cluster sizes: size $1$ and size $n^\alpha$. There are $n/2$ clusters of the first type and $n^{1-\alpha}/2$ clusters of the second type, so the total number of clusters is
> $$
>     G = \frac{n}{2} + \frac{n^{1-\alpha}}{2} \asymp n.
> $$
>     Within each cluster, assume the individual influence functions and their estimates are all identical with unit variance. Then
> $$
>     \Omega_n = \frac{n + n^{2\alpha} \cdot n^{1-\alpha}}{2n} \asymp n^{\alpha},
> $$
>     and the resulting convergence rate is
> $$
>     \sqrt{\frac{\Omega_n}{n}} \asymp n^{-(1-\alpha)/2},
> $$
>     which is slower than both $\sqrt{n}$ and $\sqrt{G}$, since $G \asymp n$. The corresponding rate condition for nuisance estimation is
> $$
>     \|\hat{\mu} - \mu\|  \|\hat{\pi} - \pi\| = o_{\mathbb{P}}\left(n^{-(1-\alpha)/2}\right), \quad \|\hat{\varphi} - \varphi\| = o_{\mathbb{P}}(1).
> $$
>     This example highlights the importance of accounting for heterogeneous cluster sizes. Although the total number of clusters $G$ is large and of the same order as $n$, the convergence rate is driven by the relatively small number of large clusters, within which the correlation is perfect."

---

### Official Review · Reviewer_HHgW · 2025-07-07

**Clarity:** 3
**Significance:** 2
**Originality:** 3
**Rating:** 4
**Confidence:** 2

**Summary:**

This paper studies the problem of estimating the average response under cluster dependence and missing outcome data, with an application to human evaluations of language models. The authors adopt the doubly robust estimation framework and extend existing theory to allow for arbitrarily sized clusters and within-cluster dependence. They prove asymptotic normality under general conditions and propose appropriate variance estimators, including cluster-robust and bootstrap methods. The methods are validated through simulation and applied to a real-world dataset from OpenAssistant Conversations dataset.

**Questions:**

1. How does your method compare empirically to outcome-only or IPW-only estimators? Including these as baselines would help contextualize the benefit of the doubly robust approach.

2. Is your variance estimator (Eq. 4) sensitive to very small cluster sizes? Are there any practical recommendations on the minimal cluster size for reliable inference?

**Ethical Concerns:**

["NO or VERY MINOR ethics concerns only"]

**Final Justification:**

After rebuttal, my concerns are properly addressed. Based on the paper quality and overall concerns, I will keep my score unchanged which lean to accept.

**Limitations:**

Yes

**Quality:**

3

**Strengths And Weaknesses:**

**Strengths**

1. The paper addresses a practically important issue in modern applications, especially in human-AI interaction scenarios such as LLM evaluation, where responses are often missing and clustered by user or conversation.

2.  It extends the asymptotic properties of doubly robust estimators under clustered sampling without requiring bounded cluster sizes, which generalizes existing work.

3. The experimental results on public dataset demonstrate the effectiveness of the proposed method.

**Weaknesses**

1. While the theoretical extension is technically solid, the core methodology is based on standard doubly robust estimation. There is no significant algorithmic innovation beyond existing frameworks.

2. The experimental section is not extensive and somewhat limited. Only one real dataset is used, with no comparisons against other estimators (e.g., outcome-only, IPW-only, or naive methods); No sensitivity analyses are conducted (e.g., varying cluster size, degree of within-cluster correlation, or misspecification scenarios).

3. The paper is dense and difficult to follow in places, especially in the theoretical sections. For example, the modeling of historical summaries (Sgt) in the sequential setting is not clearly explained or motivated, and implementation details (e.g., model architectures, embedding strategies) are sparse in the real-data section.

---

> ### Author Rebuttal · Authors · 2025-07-29
>
> We thank the reviewer for the thoughtful and constructive comments, insightful summary of our key contributions, and positive feedback on the clarity, organization, and significance of our work. These suggestions have been valuable in improving the paper. Our responses to the weaknesses and questions are as follows:
>
> **Weaknesses:**
> 1. We acknowledge that the doubly robust estimator itself is not new, and Section 3 focuses on examining its theoretical properties in clustered settings. However, a key contribution of this work also lies in Section 4, where we formalize the idea of incorporating summaries of historical information into the estimation procedure in the sequential setting—a novel approach with important applications in multiturn LLM evaluation.
> 2. See our response to Q1 below for details on additional simulations.
> 3. We have added the following discussion on the historical summaries $S_{gt}$ after Assumption 3 to help readers better understand their motivation and how they can be selected in practice:
>
> "The variable $S_{gt}$ can be viewed as a sufficient summary of the historical information up to time $t$. In practice, the choice of $S_{gt}$ often requires domain knowledge. Common choices include average or cumulative measures of past information. For example, in mobile health studies, a user wears a fitness tracker that collects data daily. The device may fail to record data at time $t$ due to battery depletion, which depends on historical usage; in this case, $S_{gt}$ could represent the cumulative device usage over the past few days. In educational testing or tutoring systems, whether a student attempts a question at time $t$ may depend on the cumulative difficulty or frustration from earlier interactions. Here, $S_{gt}$ might be defined as the number of incorrect attempts or a difficulty-adjusted score accumulated up to time $t$. In the LLM evaluation setting, whether users provide feedback may depend on their prior interactions with the system. Accordingly, $S_{gt}$ can be constructed as the embedding of the concatenated conversation history $\overline{W}_{gt}$  up to time $t$, or from the most recent $d$ conversations. These embeddings remain of fixed length regardless of the length of the conversation history."
>
> **Questions:**
> 1. We have included additional simulation studies in the appendix to compare the regression estimator, IPW estimator, and DR estimator in the clustered setting under potential model misspecification as follows:
>
> "In this section, we present additional simulation results for the plug-in (regression) estimator
> $$
> \hat{\theta}_{\mathrm{OR}} = \frac{1}{n} \sum\_{g=1}^G \sum\_{i=1}^{n\_g} \hat{\mu}(X\_g, W\_{gi}),
> $$
>
> the IPW estimator
> $$
> \hat{\theta}_{\text{IPW}} = \frac{1}{n} \sum\_{g=1}^G \sum\_{i=1}^{n\_g} \frac{R\_{gi}Y\_{gi}}{\hat{\pi}(X\_g, W\_{gi})},
> $$
> and the doubly robust estimator in equation (2). Our focus is on evaluating their performance under model misspecification in the presence of clustered data.
>
> The data-generating process follows the homogeneous sampling setup described in Appendix C, with the regression function and propensity score given by
> $$
> \mu(X\_g, W\_{gi}) = -X\_g + W\_{gi}^2, \quad \pi(X\_g, W\_{gi}) = \text{logistic}(X\_g + 0.5W\_{gi}^2).
> $$
> The true average outcome is $ \theta = 5 $. To assess estimator performance under misspecification, we consider scenarios in which $ \mu $ and/or $ \pi $ are misspecified by modeling the quadratic term in $ W\_{gi} $ as linear.
> We generate samples of size $ n= 1000, 10000 $ with varying cluster sizes, apply each estimator under different model specifications, and compute the mean squared error (MSE). The results are summarized in the following table.
>
> **Table: Mean squared error (MSE) of regression, IPW, and DR estimators under potential nuisance misspecification with sample size $n = 10000$, $n_g = 100$.**
>
> | Scenario                        | Regression Estimator | IPW Estimator | DR Estimator |
> |--------------------------------|----------------------|---------------|--------------|
> | $\mu$ correct, $\pi$ correct   | 0.0562               | 0.0561        | 0.0563       |
> | $\mu$ correct, $\pi$ wrong     | 0.0563               | 2.7023        | 0.0563       |
> | $\mu$ wrong, $\pi$ correct     | 2.0356               | 0.0563        | 0.0572       |
> | $\mu$ wrong, $\pi$ wrong       | 2.0603               | 2.6896        | 2.5765       |
>
> The conclusions in this clustered setting are similar to those in the classical i.i.d. setting. When the outcome model $ \mu $ is misspecified, the plug-in (regression) estimator $ \hat{\theta}\_{\text{OR}} $ becomes inconsistent. Similarly, the consistency of the IPW estimator $ \hat{\theta}\_{\text{IPW}} $ relies on correct specification of the propensity score $ \pi $. In contrast, the doubly robust estimator $ \hat{\theta}\_{\text{DR}} $, which models both the outcome and the missingness mechanism, remains consistent as long as either $ \mu $ or $ \pi $ is correctly specified. This aligns with the theoretical guarantees established in Theorem 1."
>
> We also include two additional tables with different sample and cluster sizes. The results are similar to the table presented above.
>
> 2. Our variance estimator in equation (4) is consistent under the assumptions stated in Theorem 1. In particular, the required scaling condition on cluster sizes is the same as in equation (3). While we allow cluster sizes to be unbounded, they need not be; the condition in (3) also accommodates small or bounded cluster sizes. Intuitively, although it may be difficult to estimate variance reliably for clusters with small sizes, their contribution to the overall variation is also small and becomes asymptotically negligible due to averaging the influence functions when calculating the DR estimator. Hence the variance estimator (4) is not sensitive to small cluster sizes. We have added the following discussion after equation (4) to clarify the consistency of the variance estimator:
>
> "Under the conditions of Theorem 1, we can show that $\hat{\Omega}_n$ is a consistent estimator of $\Omega_n$ in the sense that $\hat{\Omega}_n/\Omega_n \stackrel{P}{\rightarrow} 1$. Therefore, $\hat{\Omega}\_n / n$ can be used as a cluster-robust variance estimator for $\hat{\theta}\_{\text{DR}}$ to perform statistical inference."

---

> > ### Comment · Reviewer_HHgW · 2025-08-07
> >
> > Thanks for your response which has addressed my concerns. As my score already leans toward acceptance, I will keep it upchanged.

---

### Comment · Area_Chair_CuQ5 · 2025-08-06
**Action Required: Engage in Author-Reviewer Discussions (Extended to Aug 8 11:59 PM AoE)**

Dear Reviewers,

You must engage in discussions with authors and respond to their rebuttals before submitting your "Mandatory Acknowledgement" - simply clicking the button without participating is not permitted. The discussion period has been extended until August 8, 11:59 PM AoE, so please read rebuttals and tell authors whether they've resolved your concerns (also could ask questions if necessary). Non-participating reviewers will be flagged under the responsible reviewing initiative.

Best regards,
AC

---

### Note · Authors · 2025-08-12

**Dear AC and Reviewers,**

Thank you for your efforts in providing suggestions and improving our paper throughout this process. To summarize, in our rebuttal, we addressed the main concerns by:
- Providing additional simulation results under model misspecification, illustrating the robustness of our estimator in the clustered setting;
- Adding an example on heterogeneous cluster sizes and their effect on convergence rates;
- Expanding the discussion on the choice of historical summary variables in practice.

In the revised version, we also highlight our contributions:
- **For the causal inference and missing data literature:** we provide theoretical guarantees for doubly robust estimation in clustered settings, especially for cases with unbounded cluster sizes.
- **For the LLM evaluation literature:** we propose methods to adjust for missing human evaluation data, which allows for leveraging summaries of historical interactions and accounting for dependence in multi-turn evaluations.

Our work is both theoretical and practical, providing a novel connection between causal inference and LLM evaluation, two active areas in machine learning, and suggesting promising directions for future research, such as using balancing weights for stability and transporting evaluations across systems.

Thank you again for your consideration.

Best regards,
Authors of Submission 18603

---

### Decision · Program_Chairs · 2025-09-17

**Decision:**

Accept (poster)

**Comment:**

This paper extends doubly robust estimation to handle missing outcomes in clustered data with potentially unbounded cluster sizes. The theoretical contribution is proving asymptotic normality for the doubly robust estimator under cluster dependence without requiring bounded cluster sizes, generalizing existing work. The authors also introduce summary variables to handle sequential dependence within clusters.

The theoretical extension addresses a gap in existing works by allowing cluster sizes to grow with sample size while maintaining asymptotic properties (Reviewer XGKK, Reviewer 564S). It tackles a practical problem where clustered missing data is common (Reviewer HHgW, Reviewer XGKK), and the paper proves rate-adaptive properties in clustered settings (Reviewer 564S). The incorporation of historical summary variables for handling temporal dependence is interesting and well-motivated for sequential interactions (Reviewer XGKK, Reviewer 564S).

The contribution is theoretical rather than methodological, as the core doubly robust estimator is standard and previous work has studied missing data in clustered settings (Reviewer HHgW, Reviewer 564S). The experimental evaluation is somehow limited, using only one real dataset with simulated missing data scenario rather than real cases, and lacks comparisons with alternative methods like multiple imputation (Reviewer HHgW, Reviewer 564S). The strong assumption might not work well when missing is not at random scenarios, with limited exploration of robustness (Reviewer 564S).